# Mitochondrial ClpX activates an essential biosynthetic enzyme through partial unfolding

**Julia R Kardon[1,2,3]\*, Jamie A Moroco[4†], John R Engen[4], Tania A Baker[2,3]\***

[1]Department of Biochemistry, Brandeis University, Waltham, United States; [2]Department of Biology, Massachusetts Institute of Technology, Cambridge, United States; [3]Howard Hughes Medical Institute, Massachusetts Institute of Technology, Cambridge, United States; [4]Department of Chemistry and Chemical Biology, Northeastern University, Boston, United States

**Abstract** Mitochondria control the activity, quality, and lifetime of their proteins with an autonomous system of chaperones, but the signals that direct substrate-chaperone interactions and outcomes are poorly understood. We previously discovered that the mitochondrial AAA+ protein unfoldase ClpX (mtClpX) activates the initiating enzyme for heme biosynthesis, 5-aminolevulinic acid synthase (ALAS), by promoting cofactor incorporation. Here, we ask how mtClpX accomplishes this activation. Using *S. cerevisiae* proteins, we identified sequence and structural features within ALAS that position mtClpX and provide it with a grip for acting on ALAS. Observation of ALAS undergoing remodeling by mtClpX revealed that unfolding is limited to a region extending from the mtClpX-binding site to the active site. Unfolding along this path is required for mtClpX to gate cofactor binding to ALAS. This targeted unfolding contrasts with the global unfolding canonically executed by ClpX homologs and provides insight into how substrate-chaperone interactions direct the outcome of remodeling.

**\*For correspondence:**
kardon@brandeis.edu (JRK);
tabaker@mit.edu (TAB)

**Present address:** [†]Broad Institute, Cambridge, United States

**Competing interests:** The authors declare that no competing interests exist.

## Introduction

AAA+ protein unfoldases use the energy of ATP hydrolysis to pull apart the structures of their substrate proteins. In general, this is accomplished by gripping the substrate polypeptide with aromatic side chains of loops (the highly-conserved pore-1 loops) that protrude into the central pore of the unfoldase hexamer. ATP hydrolysis drives conformational changes in the hexamer that pivot these loops downward, thus applying a mechanical force from the site of engagement on the substrate, eventually unfolding it and translocating the unfolded polypeptide (*Sauer and Baker, 2011*).

Although some AAA+ unfoldases are specialized for non-proteolytic protein unfolding, this activity has been best characterized as part of protein degradation, in which the unfolded polypeptide is directly translocated into the proteolytic chamber of a bound peptidase. This fundamental unfoldase-peptidase architecture and mechanism is used by eukaryotic and archaeal proteasomes and by several unfoldase-peptidase complexes shared among bacteria, mitochondria, peroxisomes, and chloroplasts (*Sauer and Baker, 2011*; *Saibil, 2013*; *Quirós et al., 2015*). One such bacterial unfoldase, ClpX, with its partner peptidase ClpP, is particularly specialized for regulatory degradation of substrates rather than for protein quality control, conditionally selecting a varied repertoire of substrates to execute stress responses and cell fate decisions (*Gottesman, 2003*). Accumulating evidence indicates that mitochondrial ClpX (mtClpX) similarly acts in a regulatory capacity, although only a few substrates of mtClpX have been verified and the regulatory consequences of its actions on mitochondrial physiology are largely undetermined (*Kardon et al., 2015*; *Szczepanowska et al., 2016*; *Haynes et al., 2010*; *Matsushima et al., 2017*; *Kasashima et al., 2012*; *Al-Furoukh et al.,*

*2014*). How mtClpX recognizes its substrates also has not been characterized, although its sequence preferences for substrate recognition clearly diverge from those of bacterial ClpX (*Martin et al., 2008*).

We previously discovered that mtClpX promotes heme biosynthesis by acting on the first enzyme in the pathway, 5-aminolevulinate synthase (ALAS) (*Kardon et al., 2015*). mtClpX activates ALAS by accelerating incorporation of its cofactor, pyridoxal 5'-phosphate (PLP). This activation is non-proteolytic, but requires ATP hydrolysis and intact pore loops, implicating the unfoldase activity of mtClpX in this unconventional function. Here, we address how mtClpX is specifically deployed to direct activation of an enzyme. Using a peptide array of the ALAS sequence combined with protein mutagenesis and engineering, we defined a coherent mtClpX-binding site that spans the dimer interface of ALAS. This region contains separable elements that recruit and engage mtClpX to initiate unfolding. Using hydrogen-deuterium exchange coupled with mass spectrometry (HX MS), we observed that mtClpX induces exposure of a limited region of ALAS that extends from the mtClpX-binding site to the enclosed active site, thus opening access to the active site for cofactor binding. Engineered crosslinks that obstruct this path demonstrated that remodeling along this path is necessary for mtClpX to activate ALAS. Our observations describe the mechanism employed by a conserved mitochondrial unfoldase to activate an essential biosynthetic enzyme. They also provide a model for how the interactions between protein unfoldases and their substrates can generate widely divergent outcomes, from targeted unfolding and activation to complete unfolding and degradation.

## Results

### An N-terminal sequence directs mtClpX to activate ALAS

To determine how mtClpX induces ALAS to bind PLP more rapidly, we first sought to identify how mtClpX recognizes and engages ALAS. As an unbiased search strategy, we assayed binding of *S. cerevisiae* mtClpX to an array of peptides representing the linear sequence of *S. cerevisiae* ALAS. mtClpX bound six discrete peptide sequences within ALAS (*Figure 1A*, boxed in blue; control blot showing specificity in *Figure 1—figure supplement 1A*). Five of these sequences map to a structurally contiguous site in ALAS, consisting of an α-helix (α1) that is the most N-terminal structured element of ALAS, and a small region that is in direct hydrogen-bonding contact with α1 across the ALAS dimer interface (*Figure 1B*). The sixth, most C-terminal peptide maps to a separate site about 40 Å from the clustered sites. Because of the close proximity of many of these sequences to each other (several are within hydrogen-bonding distance of each other) and the large, multivalent surface which the ClpX hexamer provides for substrate interactions (~135 Å in diameter in a structure of *E. coli* ClpX [*Glynn et al., 2009*]), residues within several of these sequences could interact with mtClpX simultaneously as part of the initial encounter complex. Alternatively, elements within these sequences could provide contacts with mtClpX sequentially, perhaps even during initial gripping or unfolding of ALAS.

Because ClpX and other AAA+ unfoldases often initiate unfolding at or near a protein terminus, α1 of ALAS was an appealing candidate for an initial binding site for mtClpX. α1 is also immediately adjacent in both structure and sequence to a small β-sheet (β1–3) that abuts the active site. Unfolding initiated at α1 could restructure this region to allow PLP access to the active site. Indeed, we previously observed that β1–3 is conformationally responsive to the presence of PLP in our crystal structure of ALAS: One protomer of the ALAS holoenzyme dimer lost PLP during crystallization in PLP-free solvent and β1–3 of the unoccupied protomer could not be modeled. α1 and other mtClpX-interacting elements, however, remained ordered and in an equivalent conformation in both forms of the structure (*Brown et al., 2018*).

Our group previously crystallized ALAS that had been incubated with hydroxylamine (*Brown et al., 2018*), which cleaves the covalent PLP-lysine bond in the active site and converts PLP to a non-covalently bound and inactive PLP-oxime (*Figure 1—figure supplement 2A*). The structure of this inactive, PLP-depleted preparation contained a PLP-derived species in the active site despite gel filtration to remove free cofactor. Because our previous assays for activation of ALAS by mtClpX were performed with this hydroxylamine-treated ALAS, in this work we sought to determine whether mtClpX exchanges inactive cofactor, facilitates cofactor binding to unoccupied active sites, or both. We determined that ALAS treated with hydroxylamine followed by gel filtration remained slightly

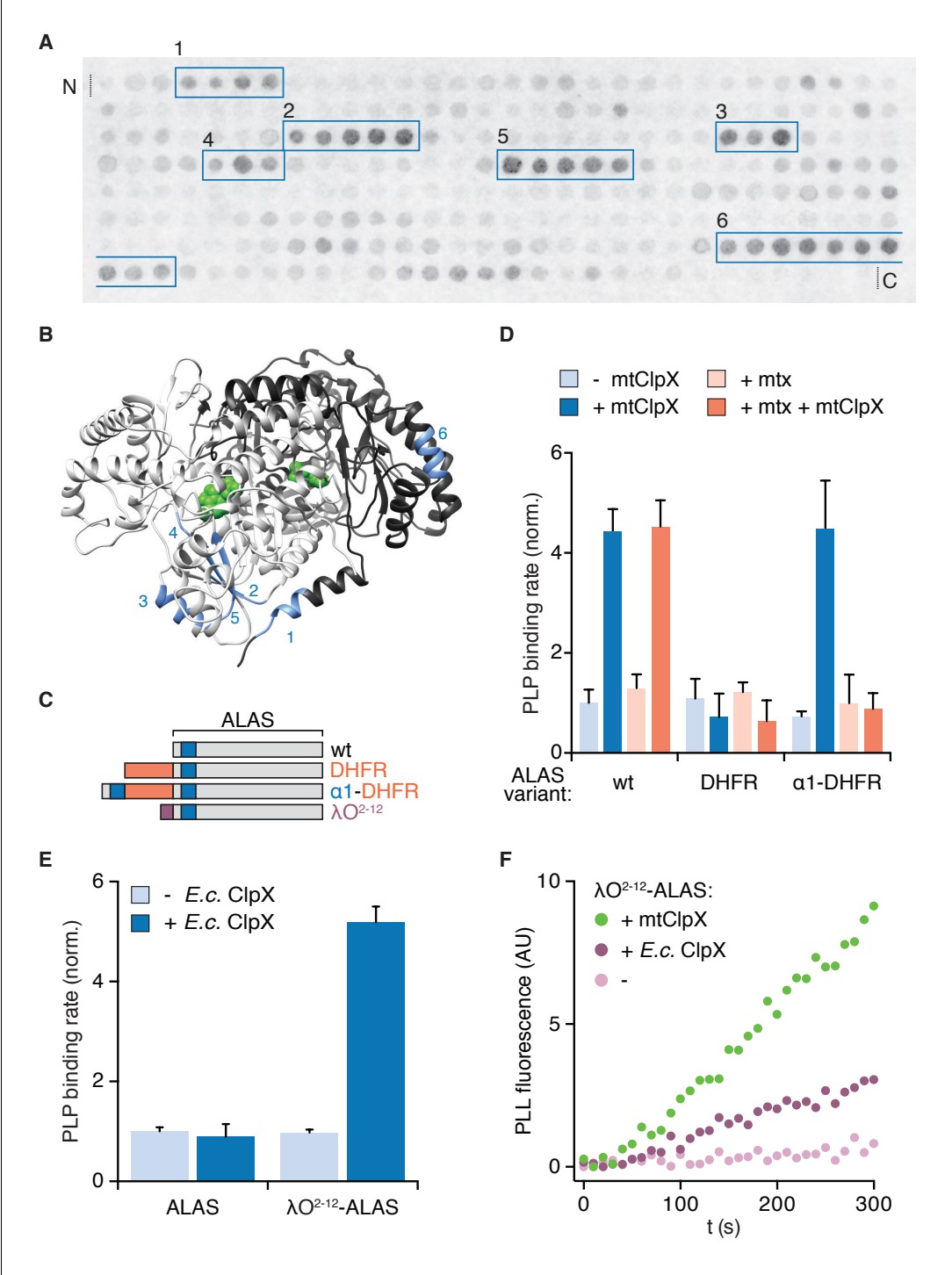

**Figure 1.** Interaction with the N-terminus of ALAS is necessary and sufficient for activation of ALAS by an unfoldase. (A) Peptide array of the ALAS sequence (58–548, sliding window of fifteen amino acids, shifted two amino acids towards the C-terminus with each spot, N- to C-terminus arrayed left-to-right, top-to-bottom) probed with mtClpX$^{E206Q}$-3xFLAG and detected by far-western blot as described in Materials and Methods. mtClpX-binding sequences are boxed in blue. See *Figure 1—figure supplement 1A* for control blot. (B) mtClpX-binding sequences identified by peptide blotting are mapped on one face of the structure of *S. cerevisiae* ALAS (PDB: 5TXR [*Brown et al., 2018*]; image created with UCSF Chimera [*Pettersen et al., 2004*]) in blue and numbered as in (A). Sequences were defined as the range between the two amino acids added at the beginning of the boxed region in (A) and the two amino acids removed after its end. PLP is depicted in green and the two protomers of ALAS are colored in light or dark gray. (C) Diagram of ALAS N-terminal variants. Blue indicates the N-terminal mtClpX-binding peptide in α1, orange indicates *M. musculus* dihydrofolate reductase (DHFR), and purple indicates a degradation tag recognized by *E. coli* ClpX (residues 2–12 of the phage λO replication protein). (D) Rate of

*Figure 1 continued on next page*

Figure 1 continued
PLP binding to ALAS and N-terminal DHFR-ALAS chimeras (5 μM monomer), ± mtClpX (2 μM hexamer), ± methotrexate (mtx) (30 μM). Reactions additionally contained 2 mM ATP, a regeneration system and 50 mM PLP (see Materials and Methods). PLP binding was monitored by fluorescence specific to protein-liganded PLP (ex. 434 nm, em. 515 nm). Rates were extracted by linear fits to values in the early linear phase and normalized to the rate for wildtype ALAS without methotrexate or mtClpX. $p < 0.001$ for suppression of mtClpX activity by DHFR fusion (DHFR-ALAS) and suppression of mtClpX activity on α1-DHFR-ALAS by methotrexate addition. (E) PLP binding to ALAS and $λO^{2-12}$-ALAS (5 μM monomer) ±E. coli ClpX (2 μM hexamer), assayed as in (C). $p < 10^{-4}$ for stimulation of PLP binding to $λO^{2-12}$-ALAS by E. coli ClpX. (F) PLP-binding fluorescence traces for $λO^{2-12}$-ALAS,±E. coli ClpX or mtClpX. Error bars represent standard deviation; $n \geq 3$. P-values were calculated using Student's t-test.
The online version of this article includes the following figure supplement(s) for figure 1:

Figure supplement 1. mtClpX-binding peptides of ALAS.
Figure supplement 2. PLP occupancy of ALAS.

more than half-occupied with a species consistent with PLP-oxime (*Figure 1—figure supplement 2A,B*). The activation of hydroxylamine-treated ALAS by mtClpX therefore represents a combination of PLP binding to unoccupied active sites and exchange of PLP-oxime for PLP. Both events are likely useful in supporting ALAS function in vivo, assisting de novo PLP binding and regenerating ALAS enzyme inactive due to a damaged cofactor (for example, PMP resulting from decarboxylation-coupled transamination of PLP, or other natural errors in catalysis [*John, 1995*]). Both events could be similarly facilitated by unfolding by mtClpX to expose the ALAS active site.

To test the importance of the putative N-terminal mtClpX binding site of ALAS, we generated variants of ALAS with dihydrofolate reductase (DHFR) appended at the ALAS N-terminus (*Figure 1C*). N-terminal fusion of DHFR with ALAS (DHFR-ALAS) blocked stimulation of PLP binding by mtClpX, but appending the N-terminal sequence of ALAS through α1 to DHFR-ALAS (α1-DHFR-ALAS) restored the ability of mtClpX to stimulate PLP binding (*Figure 1D*, dark blue bars). Thus, mtClpX recognizes the N-terminal region of ALAS to initiate activation. Upon binding its inhibitor methotrexate, DHFR becomes extremely mechanically stable and resistant to unfolding (*Rassow et al., 1989*), thus providing a conditional block to mtClpX. Addition of methotrexate to α1-DHFR-ALAS blocked its activation by mtClpX (*Figure 1D*, dark orange bars) indicating that mtClpX acting from near the N-terminus is required for ALAS activation. These data support the conclusions that mtClpX requires the α1 site to recognize ALAS and acts from this site to accelerate PLP binding.

## An unfoldase acting at the N-terminus is sufficient to activate ALAS

To test if an unfoldase acting at the N-terminus of ALAS is sufficient for activation, we attempted to direct E. coli ClpX to stimulate PLP binding. E. coli ClpX does not act on wildtype ALAS (*Figure 1E*). To direct it to ALAS, we appended a recognition sequence for E. coli ClpX ($λO^{2-12}$, from the E. coli ClpX substrate phage λO protein [*Gonciarz-Swiatek et al., 1999*]) to the N-terminus of ALAS ($λO^{2-12}$-ALAS) (*Figure 1C*). E. coli ClpX accelerated PLP binding to this fusion protein, albeit with a slower $V_{max}$ than observed with the native mtClpX-ALAS interaction (*Figure 1E*, *Figure 1—figure supplement 1B*). Therefore, an unfoldase acting from this N-terminal site is sufficient to stimulate ALAS activation. We also tested if the short, unstructured $λO^{2-12}$ tag might block mtClpX action as we observed with the folded DHFR domain did. We observed the opposite effect: mtClpX processed $λO^{2-12}$-ALAS more rapidly than wildtype ALAS by (*Figure 1F*). This increased efficiency was not based on specific interaction of mtClpX with the $λO^{2-12}$ peptide (*Figure 1—figure supplement 1C*, underlined in purple), suggesting that mtClpX instead was aided by the unstructured polypeptide extension contributed by the $λO^{2-12}$ tag.

## mtClpX relies on an unstructured N-terminal extension for rapid activation of ALAS

The observation that an unstructured extension to the N-terminus of ALAS stimulated activation of ALAS by mtClpX led us to reexamine the native N-terminus of ALAS in the mitochondrion. Many mitochondrial proteins, including ALAS, are translated with an N-terminal targeting sequence that is cleaved after import into the mitochondrial matrix. The N-terminus of the ALAS variants we used in our previous experiments (position 58 in the coding sequence) was based on an earlier proteomic

survey of mature N-termini of mitochondrial proteins in yeast (*Vögtle et al., 2009*). To specifically observe the mature N-terminus of ALAS, we performed Edman degradation on ALAS purified from yeast cell extracts. ALAS appeared as a single processed species by western blot (with possible trace uncleaved preprotein remaining, *Figure 2—figure supplement 1A*). The N-terminal sequence of this species corresponded to cleavage after residue 34 of the preprotein (Δ34-ALAS), indicating that an additional 23 amino acids beyond the previously detected species (Δ57-ALAS) are retained in the mature protein (*Figure 2A*, *Figure 2—figure supplement 1B*). One possible source for a different N-terminus is that the slower isolation procedure used to globally identify mitochondrial N-termini may have resulted in additional processing to yield Δ57-ALAS.

We tested the effect of this natural N-terminal extension of ALAS on mtClpX action and found that mtClpX activated Δ34-ALAS even more rapidly than $\lambda O^{2-12}$-ALAS (*Figure 2B*). Extension of the N-terminus of ALAS, either with its natural sequence (34-57) or with $\lambda O^{2-12}$, principally stimulated the maximal rate of activation (a nearly twenty-fold increase in $V_{max}$ by extension with the natural sequence) and modestly decreased the $K_M$ (*Figure 2B–D*). Extension of the N-terminus of ALAS also

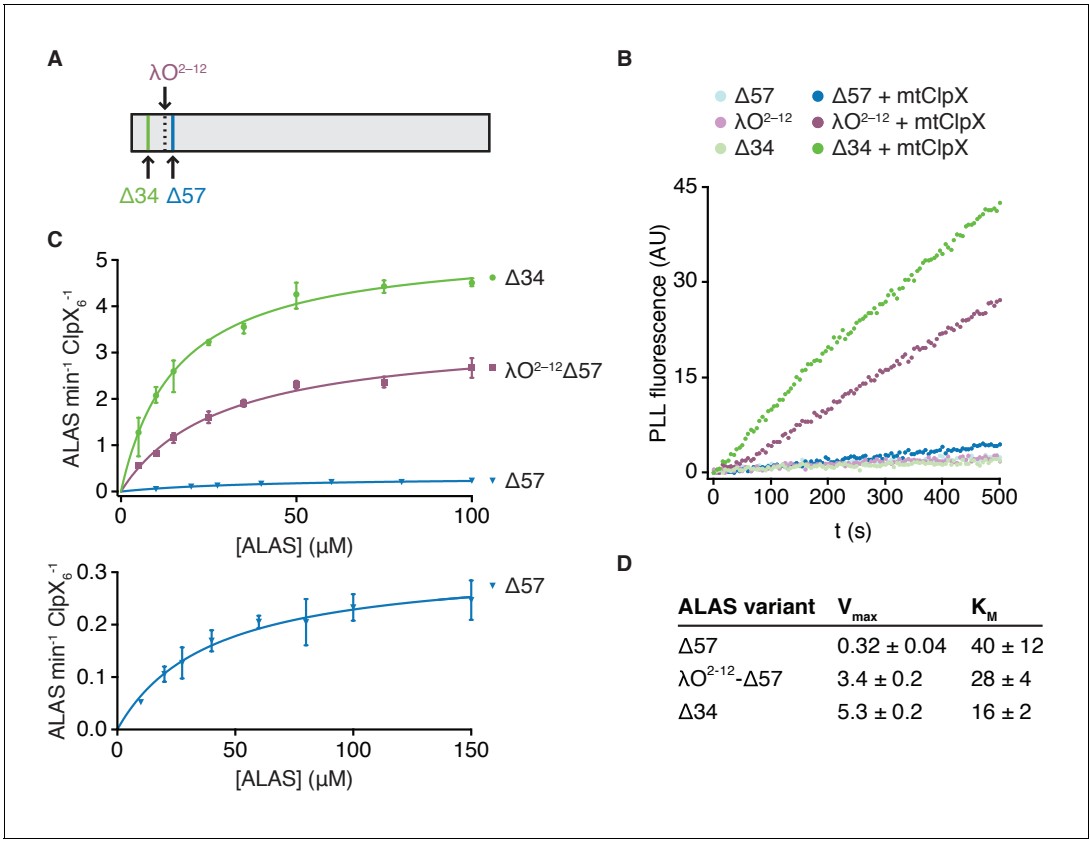

**Figure 2.** mtClpX relies on an unstructured N-terminal extension for rapid activation of ALAS. (**A**) Edman degradation of ALAS identified a single N-terminus corresponding to amino acid 35 of the preprotein. A C-terminal Myc-His$_7$ tag was integrated at the genomic locus encoding ALAS and the tagged protein was purified from yeast cell extract by Ni-NTA affinity. (**B**) Fluorescence traces representing PLP binding to 5 μM ALAS variants ± 2 μM mtClpX hexamer, monitored as in *Figure 1D*. (**C**) The rates at which mtClpX stimulated PLP binding to Δ57-ALAS (blue, n = 2), $\lambda O^{2-12}$-Δ57-ALAS (purple, n = 2), or Δ34-ALAS (green, n = 3) are plotted as a function of ALAS monomer concentration. The lower graph shows the same fitted data for Δ57-ALAS as displayed in the upper graph with a smaller y-axis scale and an additional concentration (150 μM) not monitored for the other variants. mtClpX, when included, was present at 0.5 μM hexamer ($\lambda O^{2-12}$- and Δ34-ALAS variants) or 1 μM hexamer (Δ57-ALAS). PLP was included at 150 μM. Rates were extracted by linear fits to the early phase of PLP-binding fluorescence traces; the rate of mtClpX action was determined by subtracting the ALAS-alone PLP binding rate from the mtClpX-stimulated rate and normalizing to mtClpX concentration. Curves represent fits of the Michaelis-Menten equation ($Y = V_{max}*X/(K_m + X)$ to the data. (**D**) Kinetic parameters for mtClpX action on ALAS variants, extracted from fits in (**C**). Standard error of the fit is stated.

The online version of this article includes the following figure supplement(s) for figure 2:

**Figure supplement 1.** Determination and characterization of the mature N-terminus of ALAS.

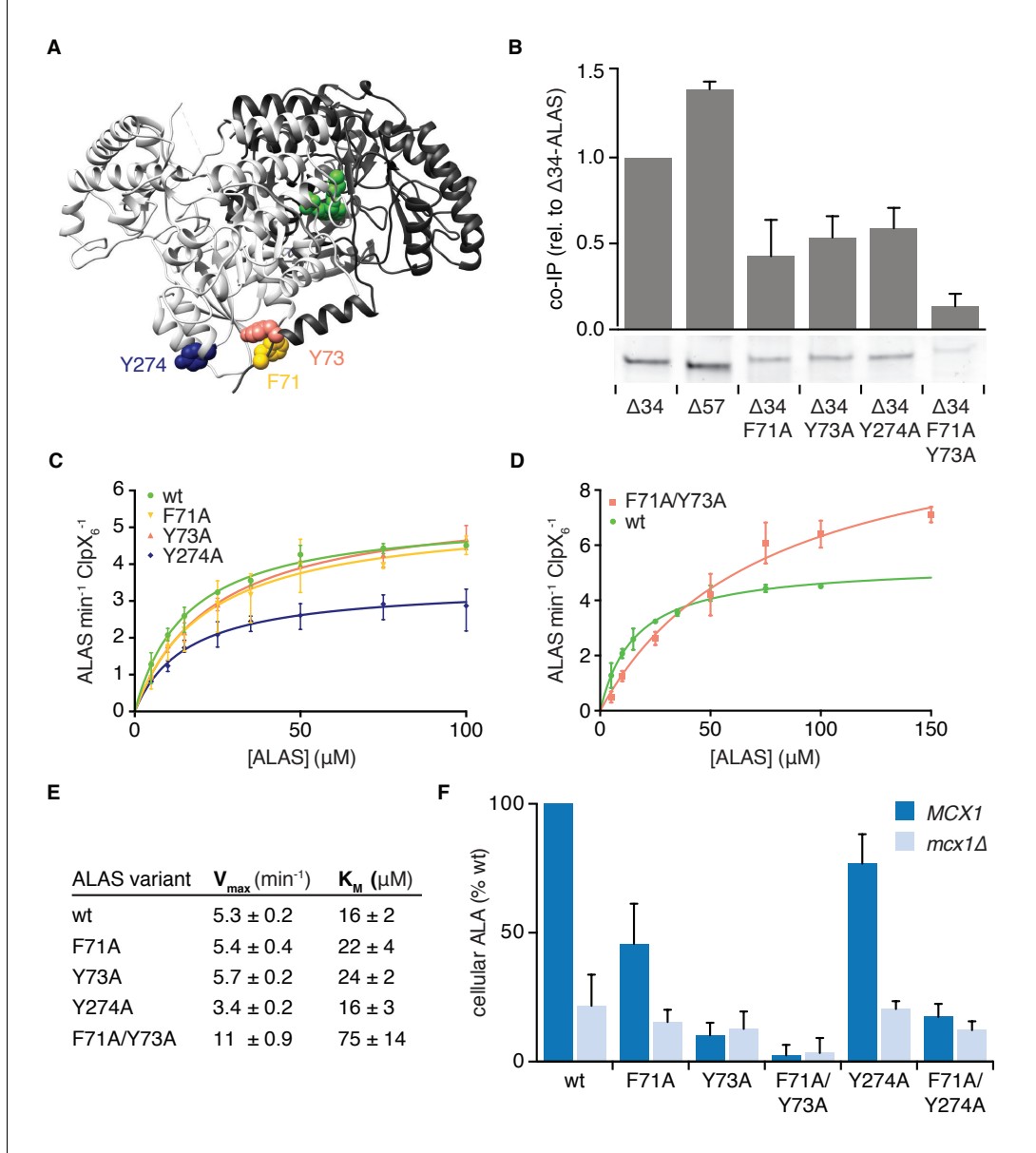

**Figure 3.** Multiple sequence-specific contacts direct mtClpX action on ALAS. (A). The position of mutations that perturb mtClpX binding are mapped on one face of the structure of ALAS (PDB: 5TXR [*Brown et al., 2018*]) (F71 in yellow, Y73 in orange, and Y274 in dark blue). PLP is depicted in green. (B) Coimmunoprecipitation of ALAS variants with mtClpX$^{E206Q}$-3xFLAG. 1 µM ALAS (monomer) and 0.5 µM mtClpX$^{EQ}$-3xFLAG (hexamer) were incubated on ice with anti-FLAG antibody-conjugated magnetic beads. Coprecipitating proteins were eluted with 3xFLAG peptide. Eluted proteins were separated by SDS-PAGE and stained with Sypro Red. The bar graph above each lane of the gel represents the average intensity of the band from three independent experiments, normalized to wildtype Δ34-ALAS; error bars represent SD. (C–D) The rates at which mtClpX stimulated PLP binding to indicated ALAS variants are plotted as a function of ALAS monomer concentration; rates were determined and fit to the Michaelis-Menten equation as in *Figure 2C*. mtClpX was present at 0.5 µM hexamer (C) or 1 µM hexamer (D). Wildtype ALAS (Δ34-ALAS) data represented in *Figure 2C* is replotted in both panels. N = 3 for all variants. (E) Chart of parameters extracted from fits in (C–D) as in *Figure 2C*. Standard error of the fit is stated. (F) The levels of ALA in extracts from yeast strains harboring the indicated mutations in ALAS (*HEM1* gene), with (*MCX1*) or without (*mcx1Δ*) the gene encoding mtClpX, were measured by colorimetric assay with modified Ehrlich's reagent. p<0.001 for reduced ALA production in *MCX1* strains by all mutations displayed in *HEM1*; p=0.05 for reduced ALA production in *mcx1Δ hem1$^{F71A/Y73A}$*, n ≥ 3 for all strains; error bars represent SD.

The online version of this article includes the following figure supplement(s) for figure 3:

**Figure supplement 1.** mtClpX-binding sequences in ALAS.

**Figure supplement 2.** Structure and function of mtClpX-interacting residues in ALAS.

reduced coprecipitation of ALAS with mtClpX (Δ34-ALAS compared to Δ57-ALAS, *Figure 3B*), suggesting that this extension may present mild steric hindrance to static binding, although it increases the overall efficiency of mtClpX activation of ALAS and may disfavor dissociation after mtClpX actively engages ALAS. The sequences of the λO$^{2\text{-}12}$ tag and of the N-terminal extension of ALAS (amino acids 35–57) are not similar (*Figure 2—figure supplement 1B*), suggesting that the role of this extension is sequence-independent. We did not detect any additional binding sequences in a peptide array that included the entire preprotein sequence of ALAS, further supporting a sequence-independent interaction (*Figure 1—figure supplement 1D*). This natural N-terminal extension of ALAS has a dynamic or disordered structure; the N-terminus of Δ34-ALAS to the beginning of α1 (residue 71) exhibited near-instantaneous completion of deuterium uptake in hydrogen-deuterium exchange experiments (*Figure 2—figure supplement 1C*; discussed further below). Therefore, mtClpX is a more potent activator of ALAS than we previously appreciated, facilitated by a flexible element at the natural mature N-terminus of ALAS.

The shorter N-terminal element we initially characterized (amino acid 58 through α1), when appended to DHFR-ALAS, was sufficient to direct mtClpX to act on this chimera with similar efficiency as on Δ57-ALAS (*Figure 1D*); we therefore considered whether the extended N-terminal element (35-α1) we identified might be sufficient to direct high-efficiency processing by mtClpX. To test this idea, we correspondingly extended the ALAS-derived N-terminus of the α1-DHFR-ALAS fusion (*Figure 1C*) and monitored mtClpX stimulation of PLP binding. PLP binding by the 35-α1-DHFR variant of ALAS was stimulated in a methotrexate-inhibited fashion by mtClpX (*Figure 2—figure supplement 1D*). With 35-α1-DHFR-ALAS, however, mtClpX acted only at a slow rate similar to that observed with Δ57-ALAS and its corresponding DHFR fusion protein, rather than the faster Δ34-ALAS rate (*Figure 1D*, *Figure 2—figure supplement 1D*). Although the poor solubility of the α1-DHFR variants of ALAS precluded determination of their $K_M$ with mtClpX, the similar and low rate of the 58-α1 and 35-α1 DHFR variants suggest that the failure of an N-terminal extension to enhance mtClpX action in this context is due to a decreased $V_{max}$. The extended 35-α1 binding site therefore is not sufficient to potentiate mtClpX action when separated by a DHFR domain from the main body of ALAS, perhaps because this geometry prevents mtClpX from forming additional contacts with ALAS that are important for efficient action.

## Multiple sequence-specific contacts direct mtClpX action on ALAS

To identify residues that recruit mtClpX and position it on ALAS for activation, we scanned the peptide sequences that bound mtClpX (*Figure 1A*) with alanine and aspartate substitutions on an additional peptide array and probed this array with mtClpX (*Figure 3—figure supplement 1*). In this array, we identified substitutions in each peptide that caused near-complete loss of mtClpX binding, with the exception of the most C-terminal peptide. Most residues sensitive to mutation were perturbed by both alanine and aspartate. Several bulky hydrophobic amino acids (leucine, phenylalanine, and tyrosine) were highly represented in positions important for mtClpX interaction. This side-chain preference differed from the characterized preference of *E. coli* ClpX, for which alanine and basic residues are often important in substrate recognition (*Flynn et al., 2003*). The mtClpX-preferred residues we observe are also notable in that they comprise the set of signal residues observed to direct mitochondrial N-end rule-based degradation (*Vögtle et al., 2009*).

To test the importance of these mtClpX-binding sequences for binding and activation of ALAS by mtClpX, we mutated several residues that were important for mtClpX interaction with ALAS-derived peptides in Δ34-ALAS. Alanine substitution at positions in α1 (F71A, Y73A, and the double substitution F71A/Y73A, *Figure 3A*) reduced coprecipitation with mtClpX (*Figure 3B*), and increased the $K_M$ for activation by mtClpX (*Figure 3C–E*), consistent with α1 functioning as a sequence-specific tag that recruits mtClpX to begin exerting force on ALAS.

The double substitution variant (ALAS$^{F71A/Y73A}$) exhibited an increased rate of spontaneous PLP binding (*Figure 3—figure supplement 2A*) and an increased $V_{max}$ for mtClpX-stimulated PLP binding (*Figure 3D,E*). These residues are positioned at the dimer interface: Y73 directly contacts the other protomer through hydrogen bonding, and F71A is positioned for a CH-pi interaction with P246 (*Figure 3—figure supplement 2B*). Although F71A and Y73A single variants retained synthase activity similar to wildtype, the double variant ALAS$^{F71A/Y73A}$ retained only ~10–20% of wildtype activity (*Figure 3—figure supplement 2C*). Removing contacts between the protomers through these mutations likely increases the solvent accessibility of the active site, thereby increasing the

spontaneous PLP-ALAS binding rate at the expense of active site function. The reduced activity of ALAS$^{F71A/Y73A}$ therefore illustrates a potential cost of a more solvent-accessible active site and suggests an advantage of transient, unfoldase-induced accessibility. Because the rate of unfolding of protein substrates by ClpX homologs is often proportional to the local mechanical stability of the substrate protein at the site of initiation of unfolding, this reduced contact may be responsible for the increased $V_{max}$ of mtClpX action on ALAS$^{F71A/Y73A}$. These residues in α1 thus participate both in mtClpX recognition and in forming stabilizing contacts across the dimer interface that suppress spontaneous PLP exchange and support ALAS activity.

In contrast to mutations in α1, a mutation at the surface of the mtClpX-interacting sequence cluster across the dimer interface from this helix, Y274A, did not perturb the $K_M$ for mtClpX activation of ALAS (although it did reduce coprecipitation [*Figure 3B*]). Instead, the Y274A mutation reduced the $V_{max}$ of mtClpX activation (*Figure 3D,E*). This finding demonstrates the importance of contacts with ALAS outside of α1 for mtClpX action and suggests that the interaction of mtClpX with Y274 and possibly other residues in this peptide cluster promote rapid processing of ALAS.

To test if the contacts between ALAS and mtClpX we identified are important for mtClpX to support ALAS activity in vivo (production of the first committed heme precursor, 5-aminolevulinic acid [ALA]), we introduced alanine mutations of the contact residues into the single genomic copy of ALAS in yeast (*HEM1*) in a wildtype or mtClpX-null (*mcx1Δ*) background. Within α1, F71A reduced and Y73A abolished mtClpX enhancement of ALA levels in vivo (*Figure 3F*). Combining both mutations (*hem1$^{F71A/Y73A}$*) reduced ALA levels further, in agreement with the low activity of the corresponding protein variant in vitro. The abundance of ALAS protein with this double mutation was also reduced (*Figure 3—figure supplement 2D*). Deletion of mtClpX in this strain caused no further reduction in ALA (*Figure 3F*). Mutation of the trans-protomer contact for mtClpX (*hem1$^{Y274A}$*) modestly reduced mtClpX-dependent ALA levels in vivo. Combined mutation of this contact and of one mtClpX α1 contact (*hem1$^{F71A/Y274A}$*) abolished mtClpX enhancement of ALA levels. The concordance between the effect of mutations in ALAS on its activation by mtClpX in vitro and mtClpX-dependent ALA production in vivo support the compound mtClpX-binding site spanned by these mutations as the physiological site at which mtClpX recognizes and begins acting on ALAS to promote ALAS activation and thus heme biosynthesis.

## mtClpX unfolds a defined region of ALAS that gates the active site

How does the mtClpX unfoldase remodel and activate ALAS, initiating from its composite recognition site? To directly address this question, we used hydrogen-deuterium exchange (HX) coupled with mass spectrometry (MS) to monitor conformational changes in PLP-depleted Δ34-ALAS undergoing remodeling by mtClpX. ALAS conformation overall was very stable in the absence of mtClpX. Most of ALAS exhibited either no measurable deuteration or gradual deuteration over the 60 min observation period, characteristic of a tightly folded protein (with the exception of fast, immediately saturated exchange at the extreme N- and C-termini) (*Figure 2—figure supplement 1C* and *Supplementary file 1*).

In the presence of mtClpX, deuteration was increased in defined, localized regions of ALAS over time (*Figure 4*; *Supplementary file 1*), strongly supporting the conclusion that mtClpX enacts limited, specific conformational change(s) rather than globally unfolding ALAS. mtClpX-induced deuterium uptake was localized to α1, the β sheet that connects α1 to the active site (β1–3) and the following active-site loop and α2, and several non-contiguous peptides surrounding the active site of ALAS (*Figure 4*). The mass spectra of most of these peptides existed in two populations, corresponding to the undeuterated and the maximally deuterated form, with a single transition between these forms (EX1 kinetics) (*Weis et al., 2006*; *Figure 4A*, right panel, indicated with the half-life of the EX1 unfolding event, $t_{1/2}$; half-life mapped on ALAS structure in *Figure 4C*, lower panel), consistent with cooperative unfolding within the peptide. This increased exposure was ATP-dependent and thus the result of active unfolding by mtClpX (*Figure 4—figure supplement 1B* and *Supplementary file 1*).

The β-sheet that mtClpX exposes partially shields the active site and also directly contacts other active-site-proximal elements, suggesting that unfolding of this element could gate PLP access. To test whether exchange in this region was the direct result of mtClpX remodeling or due to loss of PLP or PLP-oxime, we tested the effect of mtClpX on deuterium uptake of a holoenzyme preparation of ALAS as well, and observed a nearly identical uptake profile (*Figure 4—figure*

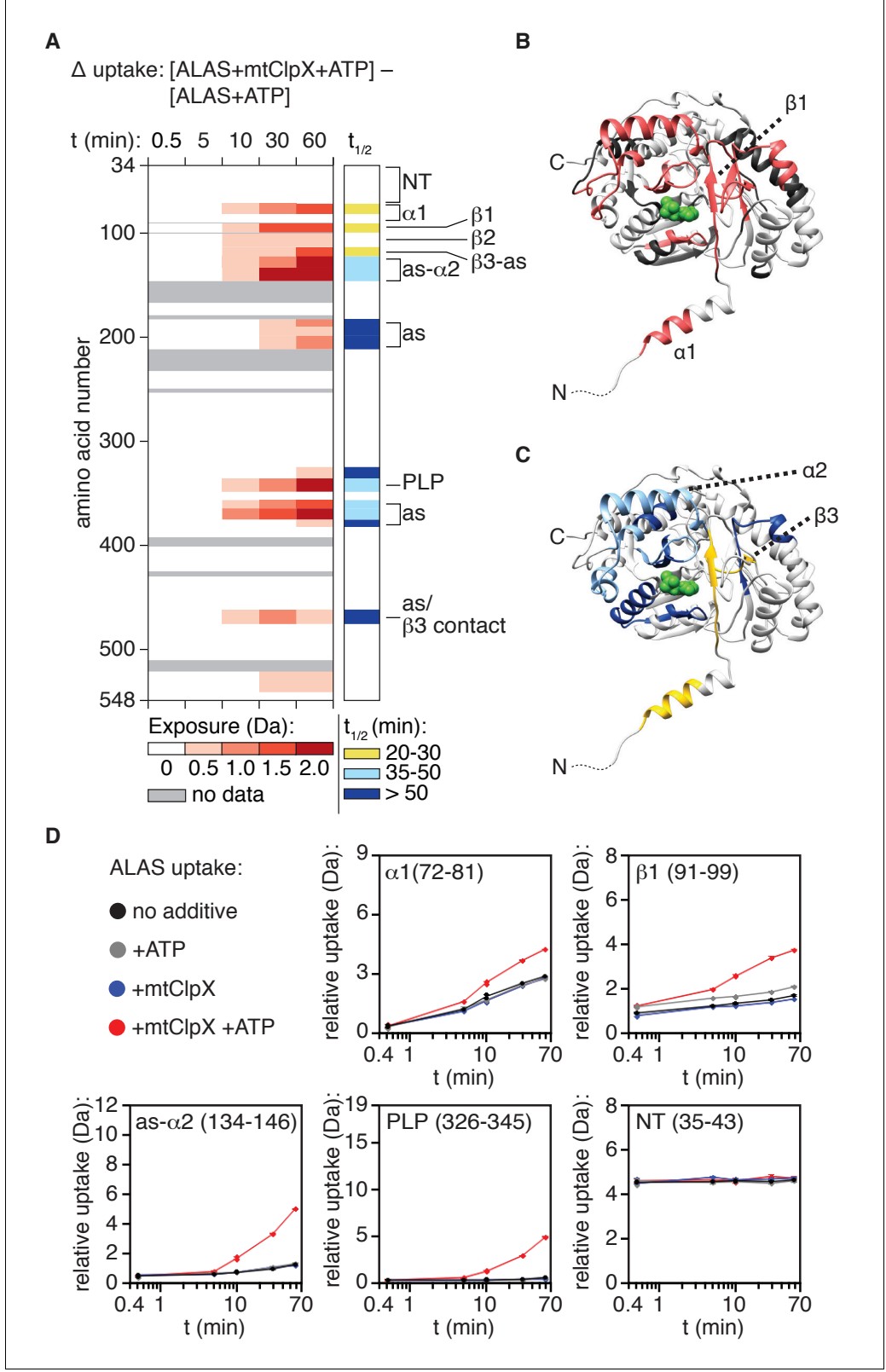

**Figure 4.** mtClpX remodels a limited region of ALAS that extends from its N-terminal binding site to the active site of ALAS. (**A**) Left panel: mtClpX-induced deuterium uptake in PLP-depleted (hydroxylamine-treated) ALAS (difference in deuterium uptake of ALAS with mtClpX and ATP present and ALAS with ATP only). A set of linear, non-overlapping peptides is shown. Legend indicates minimum value of induced deuterium uptake for each color.

*Figure 4 continued on next page*

*Figure 4 continued*

See also *Supplementary file 1* for difference maps of all peptides monitored. Peptides used in the linear map are indicated in this file. Right panel: half-life ($t_{1/2}$) of mtClpX-induced exchange for peptides in which EX1 kinetics could be clearly assigned. To the right of both panels, the correspondence of structural and functional elements of interest in ALAS with detected peptides are indicated as follows: NT: (flexible N-terminus), 35–52, 53–71; α1: 72–81, 82–89; β1: 91–99; β2, 101–113; β3-A (β3 + active site-proximal sequence): 114–122; as-α2, 125–133, 134–152; as: 184–190, 191–199, 201–212; PLP (PLP-binding active site lysine): 326–345; as: 357–363, 364–374; as/β3 contact (tertiary structure contact with β3): 461–476. (B) mtClpX-induced deuterium uptake above 0.5 Da at 10 min from (A) is mapped in salmon on the structure of ALAS (PDB: 5TXR [*Brown et al., 2018*]). One protomer of the dimer is displayed; PLP is depicted in green. (C) $t_{1/2}$ of the mtClpX-induced EX1 deuterium signatures from (A) mapped on one protomer of the ALAS dimer as in (B). Colors correspond to $t_{1/2}$ as in the right panel of (A). (D) Plots of deuterium uptake over time ± ATP, ±mtClpX for selected peptides (amino acid coordinates in parentheses) from ALAS.

The online version of this article includes the following figure supplement(s) for figure 4:

**Figure supplement 1.** Deuterium uptake difference maps.

---

*supplement 1C* and *Supplementary file 1*). Inclusion of PLP in the exchange reaction only slightly suppressed some mtClpX-induced hydrogen exchange, immediately adjacent to bound PLP (*Figure 4—figure supplement 1D,E*; see *Supplementary file 1* for data from longer peptides with clear PLP-dependent protection), indicating that remodeling by mtClpX, rather than loss of ligand, induced nearly all of the increased exposure at each element in ALAS.

Two non-homogeneous features in the linear path of mtClpX-induced hydrogen exchange suggest that mtClpX does not directly translocate the entire length of this path. First, the EX1 unfolding half-life ($t_{1/2}$) (*Engen et al., 2008*) for mtClpX-induced exchange in these elements is not uniform: exchange in α1 and β1–3 has a shorter $t_{1/2}$ than the exchange further along this path (as-α2 indicated in *Figure 1A,C*) and in the non-contiguous peptides surrounding the active site (*Figure 4A*, right panel; *Figure 4B*, lower panel). These data suggest that these regions are exposed as a secondary and/or lower-probability consequence of mtClpX remodeling of α1 and β1–3, giving rise to a longer $t_{1/2}$ for deuterium uptake. Direct unfolding of some or all of α1 and β1–3 could increase solvent exposure of the active site and might also increase the conformational dynamics of active site-proximal elements that are contacted or indirectly stabilized by α1 and β1–3. Second, a short sequence (82–89) spanning the junction between α1 and β1, elements that both exhibit fast mtClpX-induced exchange, exhibited no mtClpX-induced exchange (*Figure 5A*). A possible explanation for this observation is that this sequence in contact with mtClpX when it stops translocating, suppressing deuterium exchange even as regions further into the protein are deprotected by conformational change propagating from the putative mtClpX arrest site.

To localize the site where mtClpX is likely arrested in ALAS by an orthogonal method, we used *E. coli* ClpX with its partner protease ClpP to unfold λO$^{2-12}$-ALAS. Because ~40 amino acids of substrate polypeptide span the distance from the entrance to the ClpX pore to the active site of ClpP (*Lee et al., 2010*; *Kenniston et al., 2005*), this distance can be used to estimate the length of protein translocated by ClpX by identifying the shortest ClpXP cleavage site in a partially unfolded protein. Unlike mammalian ClpXP in vitro (*Kardon et al., 2015*), *E. coli* ClpXP could degrade ALAS if provided with an *E. coli*-specific recognition site (λO$^{2-12}$-ALAS)) (*Figure 5B*). This degradation was slowed ~4 fold when ATP was reduced to concentrations that marginally supported ATPase activity (30 μM;~20% of maximal ATPase rate) (*Figure 5B,C*, *Figure 5—figure supplement 1*). In contrast, the rate at which *E. coli* ClpX stimulated PLP binding to ALAS was only two-fold reduced when the ATP concentration was reduced from saturating to minimal (*Figure 5D*). These data suggest that *E. coli* ClpX is only sometimes impeded by the barrier in ALAS that prevents complete unfolding by mtClpX, with partial unfolding becoming more favored at low ATPase rates. No truncations of ALAS appeared during degradation, even at the low ATP concentrations that shift *E. coli* ClpX toward arrest/activation, indicating that each time *E. coli* ClpX engages ALAS, it either activates ALAS without translocating far enough into ALAS for the polypeptide to reach the active site of ClpP, or it bypasses the arrest site, continuing to complete unfolding and degradation. These data provide an upper-limit estimate of the arrest site in ALAS near the beginning of the β1 strand (~40 amino acids from the N-terminus of λO$^{2-12}$-ALAS). Furthermore, these data demonstrate that although

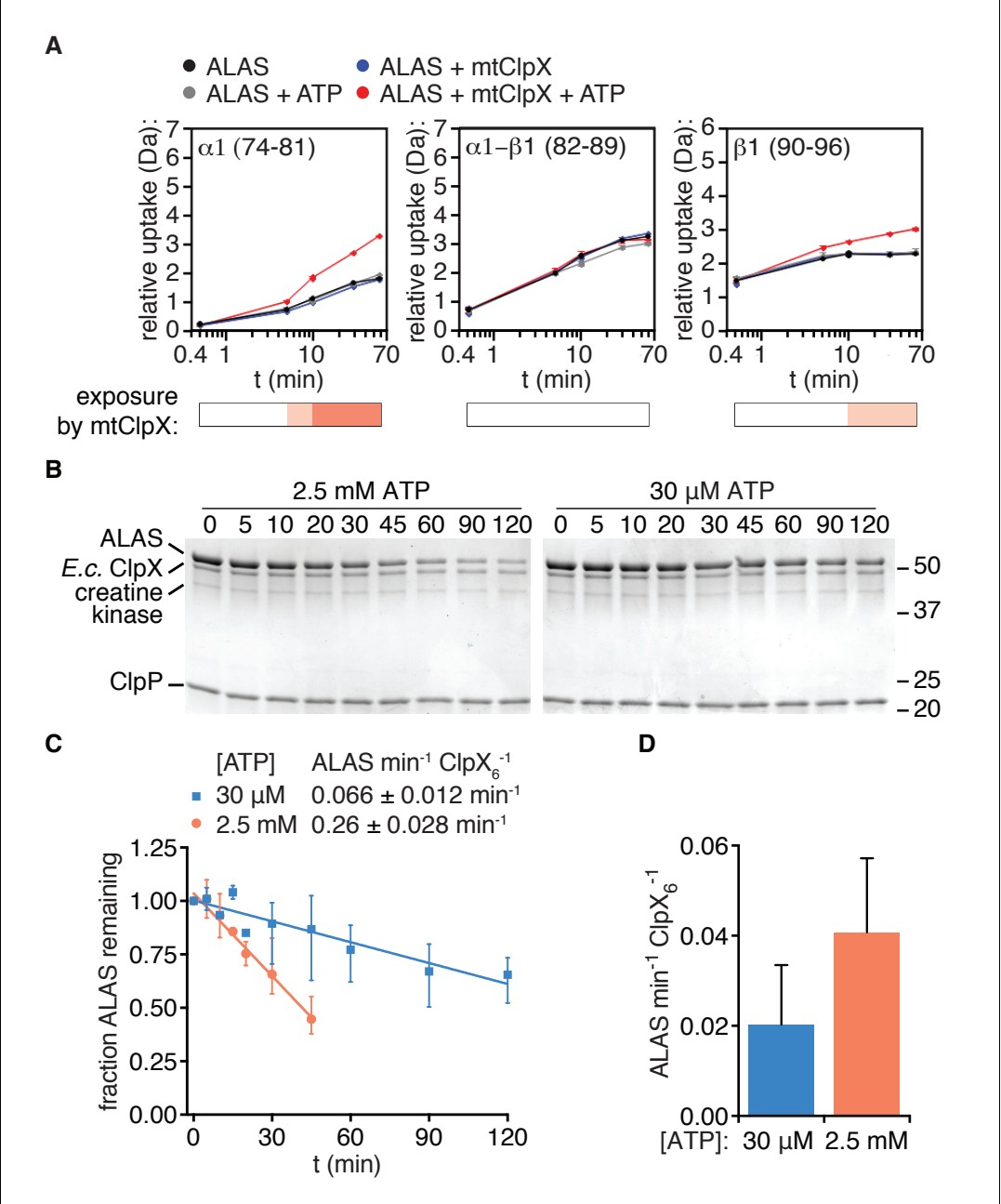

**Figure 5.** Unfolding is arrested near the N-terminus of the ordered structure of ALAS. **(A)** Deuterium uptake plots of peptides in ALAS (amino acid coordinates indicated) indicate a break in the linear path of ClpX-induced uptale. The most N-terminal alpha helix displays ClpX-induced uptake (left), the α1-β1 junction does not (middle), and the following strand β1 (right) also displays ClpX-induced uptake. Bars below the plots indicate the magnitude of ClpX-induced uptake (uptake[ALAS+ATP] – uptake [ALAS+ATP+mtClpX] as in **Figure 4** (white:≤0.5 Da; light salmon:≥0.5,≤1.0; orange:≥1.0,≤1.5. **(B)** Degradation of λO$^{2-12}$-ALAS (10 μM) was monitored in the presence of *E. coli* ClpX and ClpP (0.5 μM hexamer, 0.8 μM 14-mer, respectively) in PD150 with the indicated ATP concentration, an ATP regenerating system (5 mM creatine phosphate and 50 μg/mL creatine kinase), and 150 μM PLP at 30°C. Samples were quenched with SDS at the indicated times, separated by SDS-PAGE, stained with Coomassie R250, and imaged using a Bio-Rad ChemiDoc MP. The positions of molecular weight markers are indicated on the right in kDa. **(C)** Degradation of λO$^{2-12}$-ALAS by *E. coli* ClpXP as described in (B) was quantified from gel images using ImageJ and degradation rates were extracted by linear fit using Graphpad Prism (RRID:SCR_002798). **(D)** Stimulation of PLP binding to λO$^{2-12}$-ALAS by *E. coli* ClpX was assessed at the indicated ATP concentrations, otherwise as described in Materials and Methods.

*Figure 5 continued on next page*

*Figure 5 continued*

The online version of this article includes the following figure supplement(s) for figure 5:

**Figure supplement 1.** $K_M$ for ATP hydrolysis by *E. coli* ClpX.

appending a simple N-terminal recognition sequence to ALAS is minimally sufficient to confer partial unfolding/activation by a naïve unfoldase (here, *E. coli* ClpX) the total rate of ClpX action on ALAS (activation and degradation combined) is an order of magnitude lower than that for the native mtClpX-ALAS pair. Therefore, both high fidelity and efficiency in partial unfolding requires specific features of the cognate unfoldase and its interaction with the substrate.

Together with our mapping of mtClpX binding sites, these data indicate a path of unfolding and translocation from the N-terminus of ALAS through some portion of the first β sheet, likely arresting shortly after unfolding α1. The cooperatively-folded β sheet in ALAS (*Brown et al., 2018*) could propagate unfolding or conformational change to other regions in the protein by disrupting structural contacts and thereby triggering the exposure observed by HX MS further into ALAS. Unfolding enacted by mtClpX along this short path could thus expose the active site and gate access for PLP.

## Opening the β1–3 gate is required for ALAS activation

To test if mtClpX unfolding of ALAS along the path from the N-terminus through α1 to the following β sheet is required to stimulate cofactor binding, we designed cysteine pairs flanking this path, such that formation of a disulfide bond between these pairs would block the linear path of ClpX-driven unfolding (*Figure 6A*). No disulfide bonds between the native cysteines in ALAS were predicted; in agreement with this prediction, nearly all wildtype ALAS remained unlinked after oxidation (*Figure 6B*). Furthermore, mtClpX stimulated oxidized and reduced wildtype ALAS to bind PLP at a similar rate (*Figure 6C,D*), providing a clean background against which to observe perturbation by introduced disulfide pairs. We designed one cysteine pair (positions 68 and 243) to link the N-terminus of α1 to an adjacent site in the other protomer and a second pair (positions 88 and 427) to link the α1-β2 junction with an adjacent site in the same protomer (*Figure 6A*). ALAS[68x243] and ALAS[88x427] both migrated as apparent monomers by SDS-PAGE when reduced, but upon oxidation exhibited a near-complete shift in mobility. Oxidized ALAS[68x243] shifted to an apparent higher molecular weight, approximately twice that of an ALAS monomer, consistent with a cross-protomer disulfide bond. Oxidized ALAS[88x427] shifted to a slightly faster-migrating species, consistent with the intramolecular crosslink between these residues (*Figure 6B*). Other cysteine pairs we introduced further along the path of unfolding by mtClpX severely perturbed ALAS protein folding or stability and were thus uninterpretable in our unfolding experiments.

mtClpX action upon both oxidized variants was strongly attenuated compared to wildtype ALAS (*Figure 6C,D*). Upon reduction of the crosslink, mtClpX induced both cysteine-pair variants of ALAS to bind PLP at a similar rate as for the wildtype protein, indicating that loss of mtClpX action on these variants was specifically perturbed by disulfide bond formation (*Figure 6C,D*). Therefore, we conclude that mtClpX must unfold from the N-terminal engagement site on ALAS, extracting at least the first β strand, to activate ALAS (*Figure 6—figure supplement 1*).

## Discussion

We sought to define the mechanism by which mitochondrial ClpX deploys its activity as a protein unfoldase to activate ALAS, the initiating enzyme in heme biosynthesis. Here, we have defined sequence and structural elements that direct mtClpX to bind ALAS and unfold a substructure of the protein, gating PLP cofactor access to the active site (*Figure 6—figure supplement 1*). Our findings describe a mechanism of biogenesis and repair for an essential mitochondrial enzyme. This mechanism involves the diversion of mtClpX from the complete, global unfolding that ClpX homologs are best understood to execute. Data from our HX MS experiments and ectopically-targeted ALAS processing by *E. coli* ClpX converge to indicate that mtClpX most likely stalls near the α1-β2 junction in ALAS. This region is dissimilar to previously described unfoldase-stalling sites (*Vass and Chien, 2013*; *Tian et al., 2005*; *Zhang and Coffino, 2004*), suggesting that a greater variety of signals between unfoldases and substrates can direct alternative processing outcomes. This non-canonical

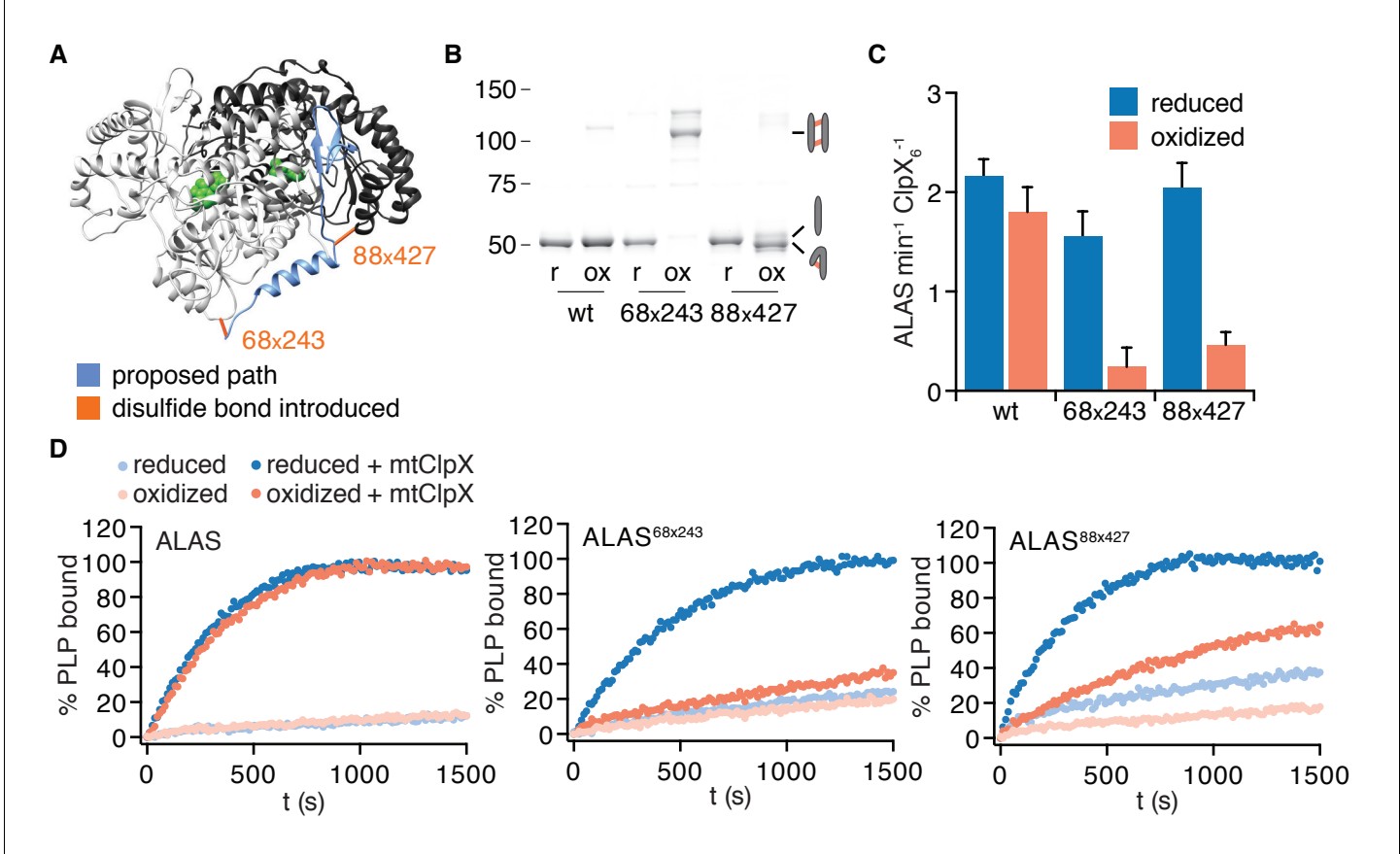

**Figure 6.** Unfolding of N-terminal secondary structure is required for mtClpX to activate ALAS. (**A**) The proposed path of mtClpX unfolding of ALAS to the active site is indicated in blue on the structure of yeast ALAS (PDB: 5TXR (*Brown et al., 2018*); image created with UCSF Chimera (*Pettersen et al., 2004*). The positions of cysteine pairs (indicated by amino acid number) introduced into ALAS are indicated by orange bonds. (**B**) Samples of ALAS cysteine variants (wt, 68–243, and 88–427) were separated by nonreducing SDS-PAGE and stained with Sypro Red, after oxidation (induced by addition of copper phenanthroline) and after reduction of oxidized samples with TCEP. Molecular weight marker (MW) sizes are indicated in kD. Cartoons indicate expected migration of species with intermolecular crosslinks (top right), uncrosslinked (middle right), and intramolecular crosslinks (bottom right). (**C**) mtClpX stimulation of PLP binding to indicated ALAS variants after oxidation and after re-reduction, monitored by fluorescence (ex. 434 nm, em. 515 nm). ALAS variants were present at 10 μM (monomer), PLP at 150 μM, ATP at 2 mM + regenerating system, and mtClpX, when included, at 0.5 μM (hexamer). Error bars represent SD from three independent experiments. p=0.002 for suppression of mtClpX action by oxidation of ALAS$^{68x243}$ and $10^{-4}$ for ALAS$^{88x427}$ (compared to reduced form of same variant). (**D**) Representative traces of PLP binding to ALAS variants. 100% PLP-bound value was set to observed plateau of PLP fluorescence.

The online version of this article includes the following figure supplement(s) for figure 6:

**Figure supplement 1.** Model for mtClpX-directed remodeling and activation of ALAS.

action of a ClpX homolog also suggests that partial unfolding may be employed to maintain and regulate a larger and more diverse set of cellular proteins than previously appreciated from studies focusing on degradation.

The elements comprising the multivalent recognition site for mtClpX on ALAS affect different parameters of mtClpX action, suggesting a model for how they cooperate to recruit mtClpX and promote unfolding of ALAS. First, a composite site that includes α1 and a region immediately across the ALAS dimer interface recruits mtClpX using sequence-specific contacts (*Figure 3*, *Figure 6—figure supplement 1*). Mutation of residues in this site decreased the apparent affinity of the ALAS-mtClpX interaction and insertion of an unfolding-resistant domain (DHFR) between this site and the body of ALAS blocked activation by mtClpX (*Figure 1*, *Figure 3*). The cross-dimer interaction site is also important for positioning mtClpX to efficiently engage and partially unfold ALAS. This importance is demonstrated by the reduction in the rate of unfoldase-mediated activation when the

unfoldase interaction with this site is perturbed in one of several ways: (i) introduction of a mutation in this site (Y274A), (ii) perturbation of its geometry relative to the N-terminal tail and α1 (as caused by insertion of DHFR), and (iii) the absence of a cognate cross-dimer site for ectopically-targeted *E. coli* ClpX, which activates ALAS at a low rate. Once positioned on ALAS, mtClpX initiates unfolding from the unstructured N-terminal tail of ALAS that precedes α1. Initiation of unfolding did not depend on the sequence of this tail; its truncation primarily decreased the $V_{max}$ of mtClpX activation rather than the strength of mtClpX-ALAS interaction. This effect suggests that the N-terminal tail of ALAS provides a readily available site for the pore loops of mtClpX to grip and initiate unfolding. This composite signal—a sequence-specific recognition site and an sequence-independent unstructured element—is similar to a two-part code observed to direct protein unfolding by the proteasomal unfoldases: ubiquitin conjugation confers recognition of a protein substrate, but a nearby unstructured element is important for efficient unfolding (*Prakash et al., 2004*).

How does unfolding by mtClpX activate ALAS? Our HX MS observations of ALAS undergoing remodeling by mtClpX identified a linear path of remodeling, starting from the N-terminal binding site for mtClpX in α1, continuing through a small β-sheet that shields PLP from solvent to the first active site loop and α2. Disulfide bonds introduced along this path blocked ALAS activation, indicating that mtClpX must unfold at least through the α1-β1 junction and extract β1 to efficiently activate ALAS, although it is possible that mtClpX directly translocates a shorter distance, or pulls from its initiation site and releases without translocating. Our experiments also localize an upper limit for how far mtClpX translocates into the structure of ALAS. The gap in the linear path of mtClpX-induced exposure at the α1-β1 junction, as well as the divergence in the halftimes of mtClpX-induced exchange within the N-terminus to the first β sheet and the halftimes of all other mtClpX-induced exchange in ALAS, suggest that the α1-β1 junction may be the approximate arrest point for mtClpX. mtClpX-induced exposure beyond this site is likely to be a secondary consequence of unfolding in the β sheet (due to solvent exposure of the active site and/or loss of cooperative tertiary and quaternary contacts [*Brown et al., 2018*]). By extracting this small portion of the otherwise intact ALAS structure, we propose that mtClpX thus opens a gate to the active site for entry of PLP and/or release of damaged PLP species (*Figure 6—figure supplement 1*).

This limited unfolding within the compact structure of ALAS poses a further mechanistic question: how is mtClpX directed to stop, mid-unfolding, and release ALAS? A few previous examples of partial unfolding by a AAA+ unfoldase as part of proteolytic processing have been characterized, including the DNA polymerase clamp loader DnaX by ClpXP in *C. crescentus* and a viral protein and the transcription factors NFκB and Ci by the 26S proteasome (*Vass and Chien, 2013*; *Tian et al., 2005*; *Zhang and Coffino, 2004*). For these substrates, as well as some engineered model substrates (*Too et al., 2013*; *Kraut et al., 2012*), the truncation point is between independent domains in the substrate protein; the unfoldase appears to be limited by encountering a mechanically stable domain and an immediately preceding sequence that stalls or releases the unfoldase. These sequences are often (but not always) glycine-rich or low-complexity (*Vass and Chien, 2013*; *Tian et al., 2005*; *Zhang and Coffino, 2004*; *Too et al., 2013*; *Kraut et al., 2012*); how these sequences induce stalling in an unfoldase is not known. The partial unfolding of ALAS we observe does not follow this formula in any obvious way. mtClpX appears to stop within a folded domain, rather than in a linker between domains. Although the natural N-terminus of ALAS is somewhat repetitive, replacement with the non-repetitive λO$^{2-12}$ sequence still supported robust stimulation of cofactor binding by mtClpX.

Several groups have recently observed mtClpX-dependent ALAS turnover under heme-replete conditions in vivo, directed by a heme-binding motif in ALAS (*Yien et al., 2017*; *Kubota et al., 2016*). These observations suggest that the block to complete unfolding by mtClpX may be conditionally released in a feedback mechanism, allowing complete unfolding and degradation (with mtClpP). The heme-binding motif falls within the flexible N-terminal extension (which diverges widely in sequence, even among vertebrates) that we here propose as a sequence-independent grip site for mtClpX. The N-terminal region of ALAS thus might serve as a bifunctional communication module for mtClpX, directing partial unfolding and activation when heme concentration is low, or ALAS degradation when heme concentration is high. Heme binding within this initiating site could reconfigure ALAS interaction with mtClpX such that it begins unfolding from a different site with a lower barrier to unfolding, or could stabilize interaction with mtClpX to disfavor release.

Unlike ClpX, the proteasomal unfoldases, and other AAA+ unfoldases that act on a wide variety of substrates (such as *E. coli* ClpX, p97/Cdc48, ClpA, and ClpB/Hsp104), some AAA+ unfoldases are dedicated modulators of one client protein. Several of these specialists, such as Rca, which de-inhibits Rubisco, or Trip13, which inactivates the mitotic checkpoint by remodeling Mad2, carry out targeted unfolding of their client that, like mtClpX with ALAS, does not appear to follow the stalling sequence/stable domain formula (*Ye et al., 2015*; *Bhat et al., 2017*; *Brulotte et al., 2017*; *Alfieri et al., 2018*). In the case of Rca, it has been proposed that its maximum unfolding force is low and thus calibrated to allow unfolding of only the C-terminal b strand of a Rubisco. In the case of TRIP13, remodeling of MAD2 destabilizes the ternary complex between TRIP13, MAD2, and their adaptor protein p31-comet, which may promote release of MAD2 by favoring TRIP13 hexamer disassembly. These adaptations seem more suited to an unfoldase with a single client protein than an unfoldase such as mtClpX that has a broader range of substrates, some of which must be completely unfolded for proteolysis.

Other substrate features and their interactions with the unfoldase may similarly direct both specialist AAA+ unfoldases and generalist AAA+ unfoldases with specialized substrates. Future probing of the arrest points of mtClpX on ALAS and these dedicated unfoldase-client pairs promises to uncover signals that allow protein unfoldases to direct varied but precise outcomes for their substrates.

# Materials and methods

## Protein purification

S.*S. cerevisiae* ALAS (Hem1) and mtClpX (Mcx1) and related variants were expressed and purified as described previously (*Kardon et al., 2015*), with modifications of reducing agent (1 mM DTT) and protease inhibitors (0.5 mM PMSF only). PLP-depleted ALAS was prepared by incubation with 5 mM hydroxylamine in 25 mM HEPES pH7.6, 100 mM KCl, 10% glycerol, and 1 mM DTT overnight on ice, followed by gel filtration (Superdex 200), concentration, and snap-freezing in liquid nitrogen. *E. coli* ClpX and ClpP were purified as described previously (*Neher et al., 2003*; *Kim et al., 2000*).

## Strain and plasmid construction

The previously described *S. cerevisiae* ALAS expression plasmid (pET28b-$H_6$-SUMO-$\Delta$57-ALAS) (*Kardon et al., 2015*) was modified by round-the-horn PCR to extend the N-terminus ($\Delta$34-ALAS, $\lambda O^{2-12}$-ALAS) and by quick-change PCR to make point mutations. For DHFR fusions of ALAS, PCR products containing the *M. musculus* DHFR coding sequence, ALAS, additional N-terminal sequences as indicated, and short flanking regions from pET28b-$H_6$-SUMO-$\Delta$57-ALAS were assembled by gap repair in pRS315 in *S. cerevisiae*, followed by subcloning of the assembled sequence into pET28b-$H_6$-SUMO. Strains used in this study are listed in *Supplementary file 1*. *S. cerevisiae* strains were constructed by homologous recombination at chromosomal loci. Strains are listed in *Supplementary file 2*.

## ALA measurement

Overnight cultures of yeast grown at 30°C in synthetic complete medium (CSM + YNB, Sunrise Science Products) + 2% glucose were used to inoculate cultures to $OD_{600}$0.1–0.15 in the same media formulation. All cultures were grown to $OD_{600}$ = 1.0–1.25 at 30°C with shaking at 220 rpm. The equivalent of 10 mL at $OD_{600}$ = 1.0 was harvested by filtration. Cell extracts were prepared and ALA was measured by colorimetric assay using modified Ehrlich's reagent as previously described (*Kardon et al., 2015*), except yeast extracts were cleared by centrifugation at 21,000 *g* rather than 2000 *g*.

## Peptide arrays

SPOT arrays of 15-amino-acid peptides, C-terminally linked to a cellulose membrane, were synthesized by standard Fmoc techniques using a ResPep SL peptide synthesizer (Intavis). Arrays were incubated with gentle agitation in methanol (5 min) followed by TBS (3 × 5 min) and then blocking solution (TBST + 5% nonfat dry milk) (2 hr). Blocked arrays were then incubated with 0.5 μM mtClpX$^{E206Q}$-3xFLAG hexamer in 25 mM HEPES pH 7.6, 100 mM KCl, 5 mM $MgCl_2$, 10% glycerol,

0.05% Triton X-100, and 5% nonfat dry milk (1 hr), washed in blocking solution (3 × 5 min), incubated with 1 µg/mL mouse anti-FLAG M2 antibody (Sigma-Aldrich Cat# F3165, RRID:AB_259529) in blocking solution (1 hr), washed in blocking solution (3 × 5 min), incubated with goat anti-mouse IgG-alkaline phosphatase conjugate (Bio-Rad Cat# 170–6520, RRID:AB_11125348) diluted 1:3000 in blocking solution (30 min), washed 5 min in block, then washed in TBST (3 × 5 min). Protein binding to the array was then imaged using ECF reagent (GE Healthcare Life Sciences) with a Typhoon FLA9500 scanner (GE Healthcare Life Sciences).

## Edman sequencing/protein isolation

*S. cerevisiae* ALAS was isolated from yeast cell extract by means of a C-terminal $His_7$ tag (strain JKY220). 1 L YPD was inoculated to $OD_{600}$ 0.05 from a saturated overnight culture and grown to $OD_{600}$ 1.0 (30°C, 220 rpm shaking). Cells were harvested by centrifugation (3500 $g$, 5 min) and washed once by centrifugation in lysis buffer (50 mM Tris, 500 mM NaCl, 10 mM imidazole, 10% glycerol). The cell pellet was snap-frozen, thawed in 15 mL lysis buffer supplemented at 1:200 with a protease inhibitor cocktail (Calbiochem EDTA-free protease inhibitor cocktail III), lysed by French press (25 kPa, Constant Systems Ltd.), supplemented with 0.5 mM PMSF, and cleared by centrifugation (30,000 $g$, 20 min, 4°C). The cleared lysate was incubated with 0.5 mL Ni-NTA agarose beads (Qiagen) for 1 hr with rotation at 4°C. The lysate-bead slurry was drained over a column and washed with 50 mL lysis buffer supplemented with 20 mM imidazole and 0.5 mM PMSF. Bound protein was eluted in lysis buffer supplemented with 250 mM imidazole, precipitated with 10% TCA, resuspended in Laemmli buffer, separated by SDS-PAGE, and transferred to Immobilon$^{PSQ}$ PVDF. After staining with Ponceau S to detect bound protein, a band corresponding to the tagged ALAS protein was excised and the N-terminal sequence was determined by Edman degradation at the Tufts University Core Facility (10 cycles, ABI 494 protein sequencer (Applied Biosystems)).

## Fluorimetry – PLP binding

PLP binding to ALAS was monitored by fluorescence (ex. 434 nm, em. 515 nm) in PD150 (25 mM HEPES pH 7.6, 150 mM KCl, 5 mM $MgCl_2$, 10% glycerol), with ALAS, mtClpX and *E. coli* ClpX concentrations as described in individual experiments. Reactions additionally contained 2 mM ATP, an ATP regenerating system (5 mM creatine phosphate and 50 µg/mL creatine kinase), and 150 µM PLP, with the exception of experiments performed with a single, 5 µM concentration of ALAS, for which PLP was included at 50 µM. Fluorescence was monitored in a 384-well plate using a SpectraMax M5 microplate reader (Molecular Devices) or in a quartz cuvette using a Photon Technology International fluorimeter.

## ALAS activity assays

ALA synthase activity of purified ALAS variant holoenzymes was monitored with 3 µM ALAS in PD150 with 100 µM succinyl-CoA and 100 mM glycine in a total volume of 60 µL. ALA content was determined using a procedure adapted from *Mauzerall and Granick (1956)* as follows. After incubation for 1 min at 30° C, 3 volumes of a 6.7% trichloroacetic acid and 13.3 mM N-ethylmaleimide solution was added to precipitate protein and quench residual DTT, respectively. After 15 min incubation on ice, solutions were centrifuged for 10 min at 21000 $g$ at 4°C. 150 µL supernatant was mixed with 50 µL 8% acetylacetone in 2 M sodium acetate and heated at 90°C for 10 min. After cooling for 5 min, 150 µL of the resulting solution was mixed with 150 µL of modified Ehrlich's reagent (20 mg/mL 4-(dimethylamino)benzaldehyde in 84% glacial acetic acid, 14% perchloric acid (70% stock) in a 96-well clear polystyrene plate and incubated for 15 min at room temperature. The absorbance at 552 nm and 650 nm was measured. ALA content was proportional to $A_{552}$ - $A_{650}$.

## Data processing and visualization

Kinetic data were plotted and analyzed by nonlinear regression using Graphpad Prism (RRID:SCR_002798). Structural data were visualized using UCSF Chimera (*Pettersen et al., 2004*, RRID:SCR_004097).

## Hydrogen-deuterium exchange and LC-MS analysis

9 µM ALAS monomer was incubated in a 15-fold excess of deuterated PD150 at 20°C with 0.5 µM mtClpX and 2 mM ATP added where indicated. At prescribed labeling times, the labeling reaction was quenched with an equal volume of ice-cold 0.3 M potassium phosphate pH 2.1. 10 µL of the quenched mixture (containing 44 picomoles ALAS) were injected onto a Waters nanoAcquity with HDX technology for online pepsin digestion and UPLC peptide separation as previously described (*Moroco et al., 2018*), with the exception that a 5–35% gradient of acetonitrile over 12 min was used. A Waters Synapt G2-Si with ion mobility enabled was used for mass analysis. Peptides were identified with triplicate undeuterated samples of ALAS alone, and in complex with mtClpX and ATP as indicated, using Waters MS$^E$ and Waters Protein Lynx Global Server (PLGS) 3.0. Peptide maps were generated and deuterium incorporation was analyzed using Waters DynamX 3.0 software. Comparison experiments were performed in duplicate, yielding the same result. One representative replicate is shown for each comparison. Only peptides found in both experiments are shown. No correction was made for back-exchange, as conditions between the compared samples were identical.

## Disulfide bond design and crosslinking

To engineer disulfide bonds in ALAS, cysteine positions were chosen using the Disulfide by Design 2.0 software (http://cptweb.cpt.wayne.edu/DbD2/index.php) (*Craig and Dombkowski, 2013*). To induce disulfide bond formation in ALAS cysteine-pair variants, proteins were buffer-exchanged (Zeba spin columns, Pierce) into PD150 and incubated with 20 µM CuSO$_4$ and 60 µM 1,10-phenanthroline for 15 min at 22°C. For analysis by SDS-PAGE, oxidized samples were quenched with 1 mM EDTA and 1 mM N-ethylmaleimide in nonreducing Laemmli buffer. For analysis of activity, samples were buffer-exchanged into PD150. To prepare reduced protein for analysis, oxidized and buffer-exchanged samples were incubated overnight with 50 mM TCEP, then exchanged into PD150.

## PLP-oxime measurements

To quantify PLP-oxime bound to ALAS, a 5 µM solution of PLP-depleted ALAS was re-supplemented with 5 mM hydroxylamine in 60 mM Na$_2$HPO$_4$ and 30 mM HCl, boiled for 5 min, cooled on ice and precipitated by addition of a 60% vol of 12% perchloric acid, followed by centrifugation (21,000 x *g*, 5 min, 4°C). The supernatant was neutralized by mixing with equal volume of 0.5 M Na$_3$PO$_4$ and fluorescence (excitation 380 nm; emission 460 nm) was monitored (adapted from a previous method for PLP quantitation [*Srivastava and Beutler, 1973*]). Fluorescence values were converted to PLP-oxime concentration by comparison with hydroxylamine-treated PLP standards.

## Acknowledgements

We thank Thomas E Wales for assistance with MS method development and thoughtful discussion, Igor Levchencko for the synthesis of peptide arrays, Michael Berne and the staff of the Tufts University Core Facility for performing Edman sequencing, and Peter Chien, Niels Bradshaw, and Steve Bell for their comments on the manuscript. JRE acknowledges a research collaboration with the Waters Corporation.

## Additional information

### Funding

| Funder | Grant reference number | Author |
|---|---|---|
| National Institute of Diabetes and Digestive and Kidney Diseases | DK115558 | Tania A Baker |
| Howard Hughes Medical Institute | | Tania A Baker |
| National Institute of Diabetes and Digestive and Kidney Diseases | DK095726 | Julia R Kardon |

| National Institute of General Medical Sciences | GM101135 | John R Engen |

The funders had no role in study design, data collection and interpretation, or the decision to submit the work for publication.

## Author contributions

Julia R Kardon, Conceptualization, Investigation, Visualization, Methodology, Writing - original draft, Writing - review and editing; Jamie A Moroco, Investigation, Visualization, Methodology, Writing - review and editing; John R Engen, Resources, Funding acquisition, Methodology, Writing - review and editing; Tania A Baker, Conceptualization, Funding acquisition, Methodology, Writing - review and editing

## Author ORCIDs

Julia R Kardon (iD) https://orcid.org/0000-0002-6621-4461
Jamie A Moroco (iD) https://orcid.org/0000-0001-8250-5923
John R Engen (iD) http://orcid.org/0000-0002-6918-9476
Tania A Baker (iD) https://orcid.org/0000-0002-0737-3411

## Decision letter and Author response

Decision letter https://doi.org/10.7554/eLife.54387.sa1
Author response https://doi.org/10.7554/eLife.54387.sa2

# Additional files

## Supplementary files

• Supplementary file 1. Deuterium uptake and difference values for all peptides monitored in ALAS.

• Supplementary file 2. *S. cerevisiae* strains used in this work. All strains were made in w303 mat a background (*MATa ade2-1 leu2-3 ura3 trp1-1 his3-11,15 can1-100 GAL psi+*).

• Transparent reporting form

## Data availability

All data generated or analysed during this study are included in the manuscript and supporting files. A source data file has been provided for Figures 4 and 5.

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
