## [Decision Letter]

**Acceptance summary:**

Using an elegant combination of peptide arrays, mutagenesis, protein engineering, and various biochemical assays, this study investigates the mechanisms and molecular determinants that control the recognition and partial mechanical unfolding of aminolevulinic acid synthase (ALAS) by mitochondrial ClpX (mtClpX) to promote PLP cofactor incorporation. The authors provide important new insights into the principles underlying non-proteolytic functions of ClpX and related motors of the AAA+ ATPase family.

**Decision letter after peer review:**

[Editors’ note: the authors submitted for reconsideration following the decision after peer review. What follows is the decision letter after the first round of review.]

Thank you for submitting your work entitled "Mitochondrial ClpX activates an essential metabolic enzyme through partial unfolding" for consideration by *eLife*, and our apologies for the delay in providing you with reviews.

Your article has been reviewed by three peer reviewers, including Andreas Martin as the Reviewing Editor and Reviewer #1, and the evaluation has been overseen by a Senior Editor.

Our decision has been reached after consultation between the reviewers. Based on these discussions and the reviews, we regret to inform you that your manuscript can not be considered for publication in *eLife* in its current form, but we encourage re-submission after the reviewers' major criticism has been addressed.

All three reviewers agreed that this manuscript describes technically challenging, sophisticated, and well executed biophysical analyses of ALAS activation by mtClpX, that the presented data are clear, elegant, and compelling, and that the manuscript is well written. However, even though this study provides new insights into mtClpX's recognition motif and initiation site for partial ALAS unfolding, there were concerns about the degree of conceptual novelty and advance compared to your previously published papers on mtClpX-mediated ALAS activation. The reviewers felt that the results did not yet converge into a coherent story, and the conclusions did not extend substantially beyond the mechanisms previously described by your group for ClpXP in general or mtClpX's action on ALAS in particular. A clearer picture of mtClpX's action, especially identifying the reasons for its arrest during ALAS unfolding and translocation, would make this a much stronger manuscript and more appropriate for publication in *eLife*. We would therefore recommend that you consider re-submission to *eLife* after those mechanistic questions have been addressed in more detail.

Major point:

What primarily limits the impact of this study is the lack of mechanistic understanding on why mtClpX stops unfolding and translocation of ALAS. As a first step, the authors should therefore attempt to identify the position of mtClpX stalling in more detail.

In their HDX-MS analyses, the authors observed a short sequence at the alpha1 / beta1 junction that exhibited no mtClpX-induced exchange, and they speculated that this may represent the stall site of mtClpX. However, it seems unlikely that the interactions of the ALAS polypeptide with pore-1 loops of a stalled yet ATP-hydrolyzing mtClpX are static and "tight" enough to prevent HD exchange to a similar extent as amide-proton protection in a hydrophobic core or when involved in H-bonds of secondary structures. Based on previous biochemical experiments and recent structures of related AAA+ motors with bound substrates, pore-loop interactions with substrate are expected to be mostly steric in nature and thus to not strongly interfere with hydrogen exchange.

With the presented data, it is difficult to predict a potential stall site for mtClpX. For instance, there is no information about cooperativity in the unfolding of ALAS' N-terminal region, and it is not ruled out that mtClpX only tugs on the N-terminus, translocates only a few residues, or dislodges alpha1 etc. HDX-MS experiments analyzing the foldedness and dynamics of N-terminally truncated ALAS variants could give insights into how disrupting individual secondary structures affects the conformation around the active site. Alternatively, and more obvious for the Baker lab, the authors could consider using ClpXP to more reliably identify the stall site through partial degradation (similar to their previous studies on partial degradation of GFP-fusion proteins by *E. coli* ClpXP in the presence of ATPgS). This information about an approximate stall site may point towards the underlying mechanism, for instance reduced grip on a low-complexity sequence etc., which can then be tested in additional experiments. The authors comment that, unlike what is proposed for the proteasomal degradation of NFkB precursors, mtClpX does not stop at an inter-domain boundary in ALAS. Perhaps there is a kinetic rather than a thermodynamic barrier to unfolding in ALAS. It would be interesting to delve more deeply into the structural features of ALAS that may block further unfolding by mtClpX, for instance using a ALAS-destabilizing mutation.

1) The authors used a peptide array to identify mtClpX-binding regions of ALAS, which are shown mapped on the ALAS dimer structure in Figure 1A. However, how do the authors envision mtClpX simultaneous interaction with regions 1, 2, 4, and 5 (and maybe 6), given their distance and differential orientation/accessibility from a particular side? Is it possible that the N-terminus of PLP-free ALAS is in a different conformation compared to the holo-enzyme? The authors mention later that the beta1-3 sheet is unresolved in the PLP-free crystal structure. This should be brought up earlier and more explicitly, as it may indeed suggest an alternative conformation for the N-terminal region of PLP-free ALAS. The HDX-MS analysis of hydroxylamine-treated, PLP-depleted ALAS (data in the Supplementary file 1) seems to show increased accessibility up to residue ~ 80 compared to holoenzyme. Either way, the authors should consider including these data for the deuterium uptake of ALAS-hxl as a main-text figure, because unfolding of apo-ALAS by mtClpX for PLP-incorporation is expected to be even more relevant than unfolding of the ALAS holoenzyme.

2) The authors use alanine and aspartate mutations in their peptide array analyses to investigate mtClpX interactions with the various N-terminal regions of ALAS, and propose a multivalent recognition site. However, it remains unclear what the contributions of regions 2-6 in the ALAS context actually are, because, according to Figure 1D, alpha1 alone placed on the N-terminus of a DHFR-ALAS fusion is as good as the delta57 N-terminus in supporting mtClpX catalyzed PLP binding. In this case, regions 2-6 are expected to not contribute to mtClpX binding due to their spatial separation from alpha1 and the translocation initiation site. The authors may consider a more detailed Michaelis-Menten analysis of the alpha1-DHFR-ALAS construct to assess the Km and Vmax effects of separating alpha1 from the rest of the ALAS N-terminus. Compared to the mutations in various binding regions, this separation of alpha1 has the advantage of preventing ClpX binding to regions 2-6 without potentially destabilizing the N-terminal region through mutations.

3) All kinetic measurements of PLP binding and the comparisons between ALAS constructs presented in Figure 1 were performed at concentrations well below the Km of mtClpX for the respective constructs, and therefore allow only tentative conclusions. For instance, that *E. coli* ClpX shows a rate of λ-O-ALAS unfolding (PLP binding) that is similar to the rate for mtClpX unfolding of wild-type ALAS may be just a coincidence (unless the Km values of mtClpX for ALAS and *E.c.* ClpX for λ-O are indeed almost the same).

According to the Michaelis-Menten analyses presented in Figure 2, the authors can produce sufficiently high concentrations of ALAS and should therefore be able to also perform the kinetic experiments presented in Figure 1 under saturating conditions and compare Vmax values.

4) The co-IP assays for mtClpX (EQ) binding various ALAS mutants (Figure 3) shows a higher amount of pulled-down delta57-ALAS compared to delta34-ALAS, despite its > 2-fold higher K_M_. The authors should try to explain this observation. Do the extra 23 residues at the N-terminus of delta34-ALAS lead to a steric hindrance and interfere with mtClpX binding? How can this be compatible with the lower K_M_ for delta34-ALAS? Could this be a consequence of the hydrolysis-dead EQ mutant, in which ATP-hydrolysis-dependent engagement of an extended tail cannot contribute to substrate affinity?

5) The authors propose a model on how heme binding to the flexible N-terminal extension of vertebrate ALAS may switch ClpX activity from partial to complete unfolding for proteolysis. However, it is unclear how effector binding to the flexible initiation region, which is threaded (and thus stripped of anything bound to it) well before ClpX reaches its stall site, could affect the outcome of ClpX translocation. Of course, the authors can only speculate, but they should try to make this proposed model on the effects of heme more consistent with their own findings of ClpX threading ALAS from the N-terminus and stalling further downstream.

6) Besides partial unfolding for PLP incorporation, mtClpX also seems to regulate ALAS protein levels. How can these two observations be reconciled? Is there an ALAS population that is fully unfolded by mtClpX? This would result in a fraction of molecules that is fully deprotected. Or does mtClpX use a different recognition motif for targeting ALAS to degradation?

7) In the second paragraph of the Discussion section, the authors state that: "its (tail) deletion increased Vmax of mtClpX rather than the avidity…".

Presumably "deletion" should be replaced with "extension", or "increased" should be changed to "decreased".

---

## [Author Response]

[Editors’ note: the authors resubmitted a revised version of the paper for consideration. What follows is the authors’ response to the first round of review.]

All three reviewers agreed that this manuscript describes technically challenging, sophisticated, and well executed biophysical analyses of ALAS activation by mtClpX, that the presented data are clear, elegant, and compelling, and that the manuscript is well written. However, even though this study provides new insights into mtClpX's recognition motif and initiation site for partial ALAS unfolding, there were concerns about the degree of conceptual novelty and advance compared to your previously published papers on mtClpX-mediated ALAS activation.

We have published two papers that address mtClpX-ALAS interaction (one identified ALAS activation by cofactor binding as an activity for mtClpX and one identified a human porphyria caused by mtClpX mutation) in addition to a paper presenting the structure of yeast ALAS. None of the previous studies address the biochemical mechanism underlying how mtClpX achieves ALAS activation; previously we established only that ATP hydrolysis by mtClpX was important, as was a mechanical element broadly required for protein unfolding by this family of enzymes (the pore-1 loop). Although these two observations supported the conclusion that the mechanochemical activity of mtClpX is involved in activation, they certainly did not demonstrate that partial unfolding, initiating from a multivalent Nterminal binding and engagement site, is the mechanism of mtClpX activation, as we demonstrate in the current manuscript. This demonstration of activation via partial unfolding is particularly relevant because activation of ALAS is an unusual activity for ClpX homologs, which have been characterized almost entirely in the context of their role within machines for degradation.

The reviewers felt that the results did not yet converge into a coherent story, and the conclusions did not extend substantially beyond the mechanisms previously described by your group for ClpXP in general or mtClpX's action on ALAS in particular. A clearer picture of mtClpX's action, especially identifying the reasons for its arrest during ALAS unfolding and translocation, would make this a much stronger manuscript and more appropriate for publication in eLife. We would therefore recommend that you consider re-submission to eLife after those mechanistic questions have been addressed in more detail.

We have performed additional experiments that further define and characterize the location of ALAS unfolding and ClpX arrest as detailed in the summary above and in specific entries below. With the added experiments helping to localize the region of mtClpX pausing, we now propose a more concrete model for how direct unfolding of the N-terminal region of ALAS by mtClpX could trigger coupled conformational changes to open the active site and facilitate PLP binding, thus activating the enzyme. These results and conclusions clearly extend our understanding of the mechanism of mtClpX activation of ALAS considerably beyond our previously published work.

Major point:What primarily limits the impact of this study is the lack of mechanistic understanding on why mtClpX stops unfolding and translocation of ALAS. As a first step, the authors should therefore attempt to identify the position of mtClpX stalling in more detail.

We attempted to observe the stall point during activation of ALAS as suggested by the reviewers by providing *E. coli* ClpXP with λO^2–12^-ALAS; based on the known distance to the proteolytic active sites, the position of ClpX on the substrate when unfolding is arrested could be estimated from the size of a truncation product that enzyme generates when stalled. Unlike mammalian ClpXP, which does not degrade ALAS on its own in vitro, *E. coli* ClpXP is able to degrade to completion a λO^2–12^-ALAS hybrid (λO^2–12^is a specific ecClpX degron) as well as activate a fraction of the enzyme for PLP binding. The rate of degradation was more severely reduced by low ATP concentration than was stimulation of PLP binding. Importantly, however, under conditions where activation was optimal, we did not observe appearance of a truncated product (Figure 5B; presented in subsection “mtClpX unfolds a defined region of ALAS that gates the active site”). These data show that *E. coli* ClpXP is only sometimes impeded in unfolding ALAS at the mtClpX stall site and that stalling compared to complete unfolding is favored at low ATP hydrolysis rates. The absence of a truncated product corresponding to where ecClpXP stalls under these conditions suggests an upper-limit estimate for the site of unfolding arrest to be near the beginning of the first b strand (~40 amino acids from the N-terminus of λO^2–12^-ALAS). These data also suggest that an additional feature(s) that is specific to the interaction between ALAS and mtClpX limits the reaction to partial unfolding with high fidelity, and this interaction/mechanism isn’t perfectly recapitulated when another unfoldase is ectopically targeted to ALAS.

In their HDX-MS analyses, the authors observed a short sequence at the alpha1 / beta1 junction that exhibited no mtClpX-induced exchange, and they speculated that this may represent the stall site of mtClpX. However, it seems unlikely that the interactions of the ALAS polypeptide with pore-1 loops of a stalled yet ATP-hydrolyzing mtClpX are static and "tight" enough to prevent HD exchange to a similar extent as amide-proton protection in a hydrophobic core or when involved in H-bonds of secondary structures. Based on previous biochemical experiments and recent structures of related AAA+ motors with bound substrates, pore-loop interactions with substrate are expected to be mostly steric in nature and thus to not strongly interfere with hydrogen exchange.

It seems reasonably likely that the arrest of mtClpX on ALAS involves a noncanonical interaction of the unfoldase (pore and/or surface) with the ALAS polypeptide and it is thus difficult to predict if this interaction would have hydrogen exchange-suppressing properties. In addition, in a recently published study of Cdc48 processing of ubiquitinated substrate (Twomey et al., 2019, 10.1126/science.aax1033), the pore loops of Cdc48 were observed to be strongly protected from exchange by statically engaged substrates (with the enzyme in the ADP-BeF-bound state). Although this effect may vary depending on the ATPase and substrate, it is possible for very tight interactions in the pore loops to dramatically slow HDX. We have edited our Discussion section in the manuscript to reflect both of these possibilities. Furthermore, as explained above, we have added a set of experiments designed to localize enzyme stall site using ecClpXP. Although this method is not a high-resolution means of mapping the location of the stalling, these data implicate the same region as that which shows no mtClpX-induced HDX. Together these two approaches strengthen the working model that stalling occurs near the alpha1-beta1 junction.

With the presented data, it is difficult to predict a potential stall site for mtClpX. For instance, there is no information about cooperativity in the unfolding of ALAS' N-terminal region, and it is not ruled out that mtClpX only tugs on the N-terminus, translocates only a few residues, or dislodges alpha1 etc. HDX-MS experiments analyzing the foldedness and dynamics of N-terminally truncated ALAS variants could give insights into how disrupting individual secondary structures affects the conformation around the active site. Alternatively, and more obvious for the Baker lab, the authors could consider using ClpXP to more reliably identify the stall site through partial degradation (similar to their previous studies on partial degradation of GFP-fusion proteins by *E. coli* ClpXP in the presence of ATPgS). This information about an approximate stall site may point towards the underlying mechanism, for instance reduced grip on a low-complexity sequence etc., which can then be tested in additional experiments.

As detailed above, we have used *E. coli* ClpXP to define the approximate stall site. We additionally tested processing of ALAS by *E. coli* ClpXP in the presence of ATPgS, which similarly slowed degradation but did not reveal a specific truncation generated when ecClpXP is working on ALAS. We consider how sequence and structural features in ALAS might affect ClpX unfolding at several points in the discussion. Our group’s previous structures of ALAS indicate that the first β sheet in ALAS behaves as a cooperative folding unit which also directly contacts the active site, supporting our model that mtClpX translocation that arrests at the beginning of this β sheet could destabilize the entire element and open the active site.

The authors comment that, unlike what is proposed for the proteasomal degradation of NFkB precursors, mtClpX does not stop at an inter-domain boundary in ALAS. Perhaps there is a kinetic rather than a thermodynamic barrier to unfolding in ALAS. It would be interesting to delve more deeply into the structural features of ALAS that may block further unfolding by mtClpX, for instance using a ALAS-destabilizing mutation.

We agree this is an interesting question. We have attempted to make mutations to perturb structural elements in ALAS that destabilize ALAS in the context of unfolding by mtClpX, but do not disrupt the basal folding and stability of ALAS. However, the mutations and truncations we tested were generally destabilizing, rendering the effect of mtClpX uninterpretable in this context. Our conclusion from these and additional experiments is that further definition of the stop-unfolding signal(s) and their mechanism of action will most likely require high-resolution structural data combined with extensive biochemical experiments, a project that is beyond the scope of this manuscript.

1) The authors used a peptide array to identify mtClpX-binding regions of ALAS, which are shown mapped on the ALAS dimer structure in Figure 1A. However, how do the authors envision mtClpX simultaneous interaction with regions 1, 2, 4, and 5 (and maybe 6), given their distance and differential orientation/accessibility from a particular side? Is it possible that the N-terminus of PLP-free ALAS is in a different conformation compared to the holo-enzyme? The authors mention later that the beta1-3 sheet is unresolved in the PLP-free crystal structure. This should be brought up earlier and more explicitly, as it may indeed suggest an alternative conformation for the N-terminal region of PLP-free ALAS. The HDX-MS analysis of hydroxylamine-treated, PLP-depleted ALAS (data in the Supplementary file 1) seems to show increased accessibility up to residue ~ 80 compared to holoenzyme. Either way, the authors should consider including these data for the deuterium uptake of ALAS-hxl as a main-text figure, because unfolding of apo-ALAS by mtClpX for PLP-incorporation is expected to be even more relevant than unfolding of the ALAS holoenzyme.

mtClpX interaction with the peptides we identified may not be simultaneous but could instead be sequential. For example: (i) these contacts could function as “enhancement tags”, such as have been previously characterized as functioning to bring the substrate near to its initiation site, raising the effective concentration of the substrate to the protein-processing core or (ii) alternatively, some of the peptide interactions could occur after initiation of unfolding perhaps to help stabilize ClpX-ALAS complexes during the initial unfolding steps. We have revised our discussion of mtClpX-ALAS binding in the text (” An N-terminal sequence directs mtClpX to activate ALAS”) to explicitly consider sequential binding. However, we also would not rule out simultaneous interaction of ClpX with several of these peptide sequences. Many AAA+ unfoldases/proteases have multiple distinct peptide-binding sites, including ecClpX. Most peptides (perhaps except 4, the most internal) could plausibly contact different parts of the mtClpX hexamer simultaneously (in structure of the *E. coli* enzyme, the hexamer has a diameter of ~135 Å). The contact outside of peptide 1/alpha1 that we are most confident in (tyrosine 274 in peptide 5, discussed further below in our response to point 2) lies immediately adjacent to peptide 1, and it is very plausible that these two binding sites function in concert (see Figure 3—figure supplement 2).

Concerning the conformation of ALAS when PLP bound or free in the crystal structures we obtained previously: although the beta1-3 sheet was unresolved when adjacent to a PLP-free active site (as described in detail in the referenced publication), peptide1/alpha1 as well as the other mtClpX-interacting peptides remained resolved and in an equivalent conformation, therefore not supporting any difference in accessibility of these peptides to mtClpX, as we have now explicitly stated in the text. We refer to these structural observations in the second paragraph of the results, immediately after a few sentences describing the first new data presented in this manuscript (the initial peptide-blot results); referring to it earlier would seem odd and out of context.

Your comments brought to our attention that our nomenclature for the ALAS protein used in the HX-MS experiments presented in Figure 4 was incomplete; the data displayed here were obtained with PLP-depleted ALAS (hxl-ALAS); the holoenyzme data is presented only in the supplemental figures (we agree – it is the less relevant species). We have changed the figure legend as well as nomenclature in the related text to make this clear. Although there are some subtle differences in the HX-MS data for mtClpX effect on hxl- and holo-ALAS (mtClpX-stimulated hxl-ALAS linear uptake plot in Figure 4 panel A, mtClpX-stimulated holo-ALAS linear uptake plot in Figure 4—figure supplement 1 panel C, full uptake data for all reproducibly-detected peptides from both conditions in supplementary file 1), they are within the noise of these measurements; we therefore concluded it is most accurate to state only that these uptake profiles are very similar.

2) The authors use alanine and aspartate mutations in their peptide array analyses to investigate mtClpX interactions with the various N-terminal regions of ALAS, and propose a multivalent recognition site. However, it remains unclear what the contributions of regions 2-6 in the ALAS context actually are, because, according to Figure 1D, alpha1 alone placed on the N-terminus of a DHFR-ALAS fusion is as good as the delta57 N-terminus in supporting mtClpX catalyzed PLP binding. In this case, regions 2-6 are expected to not contribute to mtClpX binding due to their spatial separation from alpha1 and the translocation initiation site. The authors may consider a more detailed Michaelis-Menten analysis of the alpha1-DHFR-ALAS construct to assess the Km and Vmax effects of separating alpha1 from the rest of the ALAS N-terminus. Compared to the mutations in various binding regions, this separation of alpha1 has the advantage of preventing ClpX binding to regions 2-6 without potentially destabilizing the N-terminal region through mutations.

A full Michaelis-Menten analysis of the DHFR-ALAS fusions unfortunately was not feasible; the yield and solubility of these constructs were poor. We can, however, provide two pieces of data that support the importance of regions of ALAS outside of alpha1 for interaction with mtClpX:

1) Extension of alpha1-DHFR-ALAS from the originally presented form with a short N-terminal tail (D57-alpha1-DHFR-ALAS) to a form with the full-length mature N-terminus (D34-alpha1-DHFR-ALAS) does not stimulate mtClpX action (Figure 2—figure supplement 1, related text in subsection “mtClpX relies on an unstructured N-terminal extension for rapid activation of 150 ALAS”), even though this N-terminal extension in its native context increases the maximal rate of mtClpX activation of ALAS by more than 10-fold. Therefore, although the rate of mtClpX-stimulated PLP binding for the alpha1-DHFR-ALAS variants is similar to that for D57-ALAS, it is much slower than the rate of mtClpX-stimulated PLP binding for the full-length mature form of ALAS, D34-ALAS. These data show that the N-terminal alpha1 binding site for mtClpX on ALAS is sufficient for only an attenuated rate of mtClpX activation of ALAS; therefore, it is likely that contacts with mtClpX elsewhere in ALAS are necessary to recruit and position mtClpX for maximally efficient action.

2) Mutation of a residue in region 5 in ALAS, Y274A, reduced ALAS coprecipitation with mtClpX (Figure 3B) and reduced the V_max_ of mtClpX acting on ALAS (Figure 3C,E). In cells, this mutation reduced mtClpX-dependent ALA production, modestly on its own and as a near-complete reduction in combination with the alpha1 mutation F71A (Figure 3F). We have edited the text relating to Figure 3 to explicitly highlight the support this result provides for the importance of regions other than alpha1 for directing mtClpX-ALAS interaction. In the course of these experiments, we also tested other mutations outside of alpha1 suggested by the alanine and aspartate scanning of peptides for an effect on mtClpX-ALAS activity. Although some of these other mutations did exhibit perturbed interaction and activation by mtClpX, they also exhibited strong perturbation of their enzymatic activities and basal (unstimulated by mtClpX) PLP binding (unsurprising given the high sequence conservation of ALAS, including in these regions) and thus the primary source of their defect was ambiguous and therefore not informative with respect to the reviewers’ question.

3) All kinetic measurements of PLP binding and the comparisons between ALAS constructs presented in Figure 1 were performed at concentrations well below the Km of mtClpX for the respective constructs, and therefore allow only tentative conclusions. For instance, that *E. coli* ClpX shows a rate of λ-O-ALAS unfolding (PLP binding) that is similar to the rate for mtClpX unfolding of wild-type ALAS may be just a coincidence (unless the Km values of mtClpX for ALAS and E.c. ClpX for λ-O are indeed almost the same).According to the Michaelis-Menten analyses presented in Figure 2, the authors can produce sufficiently high concentrations of ALAS and should therefore be able to also perform the kinetic experiments presented in Figure 1 under saturating conditions and compare Vmax values.

As discussed in our response to point 2, the DHFR-ALAS constructs presented in Figure 1 were insufficiently soluble to be amenable to a full kinetic analysis, but the single-concentration data do provide support for the qualitative point that mtClpX must engage the N-terminal-alpha1 feature of ALAS to unfold it with any efficiency (a point which is further supported by the effect of crosslinking this element, in Figure 6). With respect to the *E.c* ClpX-λ-O-ALAS unfolding rate, the qualitative result that transplantation of an *E. coli* ClpX recognition sequence to the N-terminus of ALAS confers some ability to stimulate PLP binding, rather than a similarity in rate, was most relevant for our conclusions. Indeed, the effect is only similar even at sub-saturating concentrations for the short form of ALAS (D57-ALAS); the longer form of ALAS that we detected (D34-ALAS) is much more rapidly acted on by mtClpX than *E.c.* ClpX (Figure 1F). We have added a Michaelis-Menten analysis of *E.c.* ClpX acting on λ-O-ALAS (Figure 1—figure supplement 1B) that demonstrates a V_max_ much lower than for the native ALAS-mtClpX pair with a slightly lower K_M_ (likely dictated largely by the λ-O sequence). We have edited the related text to reflect these points.

4) The co-IP assays for mtClpX (EQ) binding various ALAS mutants (Figure 3) shows a higher amount of pulled-down delta57-ALAS compared to delta34-ALAS, despite its > 2-fold higher K_M_. The authors should try to explain this observation. Do the extra 23 residues at the N-terminus of delta34-ALAS lead to a steric hindrance and interfere with mtClpX binding? How can this be compatible with the lower K_M_ for delta34-ALAS? Could this be a consequence of the hydrolysis-dead EQ mutant, in which ATP-hydrolysis-dependent engagement of an extended tail cannot contribute to substrate affinity?

In the static-binding regime of the co-IP experiment, we think that it is likely that the additional length of the D34-ALAS N-terminal tail presents a mild steric hindrance to forming the initial encounter complex with mtClpX, thus slightly decreasing coprecipitation. However, when mtClpX is actively remodeling ALAS, this increased tail likely helps enzyme engagement and antagonizes dissociation at later steps in the pathway to unfolding, thus leading to a modestly decreased K_M._ We have added a brief discussion of this possibility in subsection “mtClpX relies on an unstructured N-terminal extension for rapid activation of 150 ALAS”.

5) The authors propose a model on how heme binding to the flexible N-terminal extension of vertebrate ALAS may switch ClpX activity from partial to complete unfolding for proteolysis. However, it is unclear how effector binding to the flexible initiation region, which is threaded (and thus stripped of anything bound to it) well before ClpX reaches its stall site, could affect the outcome of ClpX translocation. Of course, the authors can only speculate, but they should try to make this proposed model on the effects of heme more consistent with their own findings of ClpX threading ALAS from the N-terminus and stalling further downstream.

Although the model is indeed speculative at this point, we envision several possible mechanisms by which heme binding could direct mtClpX to completely rather than partially unfold ALAS. For example, heme binding at the N-terminus could delay mtClpX dissociation and kinetically favor further unfolding, or redirect mtClpX to initiate unfolding at another site (for example, in the recently deposited human ALAS2 structure, the most C-terminal ordered portion of the protein is positioned near alpha1 and is followed by a nine-residue disordered tail). We have added a brief description of such a possible model to our Discussion section.

6) Besides partial unfolding for PLP incorporation, mtClpX also seems to regulate ALAS protein levels. How can these two observations be reconciled? Is there an ALAS population that is fully unfolded by mtClpX? This would result in a fraction of molecules that is fully deprotected. Or does mtClpX use a different recognition motif for targeting ALAS to degradation?

The limited available evidence points to heme binding to a motif with the N-terminal disordered region of ALAS as a signal that directs mtClpX-dependent degradation in cells. Under normal, low free heme concentrations in cells, correspondingly low occupancy of this motif by heme could direct only a low frequency of complete unfolding and degradation by mtClpX. We have added a more concrete discussion of a model for heme-mediated redirection of the mtClpX-ALAS interaction from partial unfolding to complete unfolding, as mentioned above.

7) In the second paragraph of the Discussion section, the authors state that: "its (tail) deletion increased Vmax of mtClpX rather than the avidity…".Presumably "deletion" should be replaced with "extension", or "increased" should be changed to "decreased".

Thank you for catching this – corrected.